# Improving semantic understanding in speech language models via brain-tuning

**Omer Moussa**[1]     **Dietrich Klakow**[2]     **Mariya Toneva**[1]

[1]Max Planck Institute for Software Systems
{omoussa, mtoneva}@mpi-sws.org

[2]Saarland University
dietrich.klakow@lsv.uni-saarland.de

## Abstract

Speech language models align with human brain responses to natural language to an impressive degree. However, current models rely heavily on low-level speech features, indicating they lack brain-relevant semantics which limits their utility as model organisms of semantic processing in the brain. In this work, we address this limitation by inducing brain-relevant bias directly into the models via fine-tuning with fMRI recordings of people listening to natural stories–a process we name *brain-tuning*. After testing it on 3 different pretrained model families, we show that brain-tuning not only improves overall alignment with new brain recordings in semantic language regions, but also reduces the reliance on low-level speech features for this alignment. Excitingly, we further show that brain-tuning leads to 1) consistent improvements in performance on semantic downstream tasks and 2) a representational space with increased semantic preference. Our results provide converging evidence, for the first time, that incorporating brain signals into the training of language models improves the models' semantic understanding. We make the code available at https://github.com/bridge-ai-neuro/brain-tuning.

## 1 Introduction

It is an exciting time for the cognitive neuroscience of language with the rise of language models which have been shown to align with (i.e. predict) brain activity evoked by natural language to impressive and unprecedented degrees (Wehbe et al., 2014; Jain & Huth, 2018; Toneva & Wehbe, 2019; Schrimpf et al., 2021; Caucheteux & King, 2022; Goldstein et al., 2022; Karamolegkou et al., 2023). Researchers aim to use language models as model organisms (Toneva, 2021) of reading and listening in the brain to learn more about the underlying information processing that leads to brain-like representations of language.

However, recent work has questioned whether current popular speech language models can fully serve this role, as their alignment with semantic brain regions was shown to be mostly due to low-level speech features, indicating that speech language models lack brain-relevant semantics (Oota et al., 2024a). Given that most large brain recording datasets are of speech-evoked language (LeBel et al., 2023; Nastase et al., 2021; Deniz et al., 2019; Momenian et al., 2024), having speech models with improved brain-relevant semantics is important to provide better model organisms for auditory language processing. The lack of brain-relevant semantics in speech models (Oota et al., 2024a) may also be related to their incomplete downstream semantic understanding. (Choi et al., 2024).

To bridge the gap between language understanding in speech models and the human brain, we propose to augment pretrained speech model training directly with brain recordings in a process we call brain-tuning (see Fig.1a for illustration of the training approach). We then evaluate the resulting brain-tuned speech models in three distinct ways (see Fig.1c for an illustration of the evaluation approach): 1) alignment with new brain recordings in semantic regions of the brain, which we expect to significantly increase if brain-tuning successfully induces brain-relevant semantics, 2) effect of low-level features, such as Tri-Phones and Articulation, on the alignment with these semantic regions, which we expect to significantly decrease if brain-tuning successfully induces brain-relevant semantics 3) downstream performance on tasks that are helped by semantic understanding, which we expect to significantly improve if the brain-relevant semantic understanding induced by the brain-tuning is also useful for downstream semantic tasks.

We brain-tune three popular speech language models using the largest available fMRI dataset, recorded when participants listened to natural stories. Across all models, we find that brain-tuning 1) significantly improves alignment with new fMRI recordings in semantic brain regions, 2) significantly reduces the impact of low-level features on this alignment, and 3) significantly improves downstream performance on tasks that are helped by semantic understanding. We show that these results hold when comparing the brain-tuned models to their pretrained counterparts, and to two additional strong baselines (i.e. brain-tuning with block-permuted fMRI data, and fine-tuning using representations from a larger speech model).

Our results provide converging evidence that augmenting speech models with brain signals from listening to natural language improves semantic understanding in speech models. Excitingly, our findings indicate for the first time that improving alignment with semantic understanding in the brain also translates to downstream gains for the models. We will make all models and code publicly available, and hope that the improved speech models our work provides will contribute to a better understanding of listening in the brain.

Our main contributions can be summarized as follows:

1. We provide an approach to fine-tune pretrained speech models using fMRI recordings of people listening to natural stories, and validate it across three popular model families.

2. We conduct extensive analyses to understand the impact of this fine-tuning on the speech model representations and behavior.

3. For the first time, we show that improving alignment with the brain has a substantial and significant downstream benefit for an AI model.

## 2 RELATED WORK

Our work is most closely related to that of Schwartz et al. (2019), who fine-tune one pretrained text-based language model (BERT (Devlin et al., 2019)) using fMRI and MEG recordings of participants reading a chapter of a book. We instead focus on speech models, validate our method across three model families, and conduct comprehensive analyses to reveal that brain-tuning improves semantic understanding in speech language models for the first time. Separately, a growing literature investigates the alignment between human brains and pretrained language models. A number of studies have shown a degree of alignment between language-evoked brain activity with text-based language models (Wehbe et al., 2014; Jain & Huth, 2018; Toneva & Wehbe, 2019; Caucheteux & King, 2022; Jat et al., 2019; Abdou et al., 2021; Schrimpf et al., 2021; Toneva et al., 2022a;b; Antonello et al., 2021; Oota et al., 2022; Merlin & Toneva, 2022; Aw & Toneva, 2023; Oota et al., 2024b; Lamarre et al., 2022; Antonello et al., 2024), and with speech-based language models (Millet et al., 2022; Vaidya et al., 2022; Tuckute et al., 2023; Oota et al., 2023; 2024a; Chen et al., 2024). Our approach of brain-tuning pretrained language models is complementary and can be used in addition to previous methods for analyzing the alignment between language models and brain activity.

## 3 METHODS

### 3.1 SPEECH LANGUAGE MODELS

We build on three popular pretrained transformer-based speech language model families: Wav2vec2.0 (Baevski et al., 2020), HuBERT (Hsu et al., 2021), and Whisper (Radford et al., 2023). We chose versions of these models that have comparable sizes ($\sim$90M parameters), the same number of encoder layers (12), and the same embedding size (768). Wav2vec2.0 and HuBERT are self-supervised models that are trained to predict representations of masked portions of the input. They both divide the input into tokens of 20ms and then use a CNN feature extractor. We use the base architectures which are trained on $\sim$960 hours of audio. Whisper, unlike Wav2Vec2.0 and HuBERT, is trained in a weakly supervised manner, using 680K hours of paired audio-text data and has an encoder-decoder architecture. Contrary to HuBERT and Wav2Vec2.0, Whisper takes a fixed 30s input and then converts it to log-mel spectrograms. We fine-tune only the Whisper encoder for two reasons: 1) to keep the model of comparable size to the other two models, and 2) since the encoder

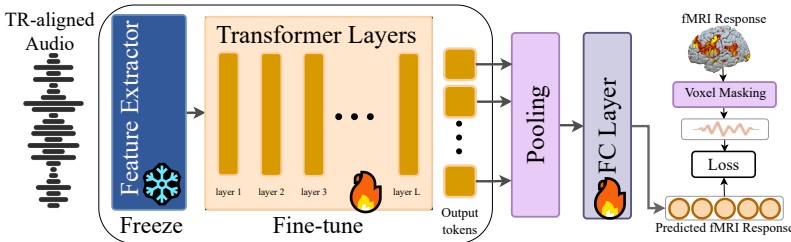

(a) Proposed brain-tuning approach

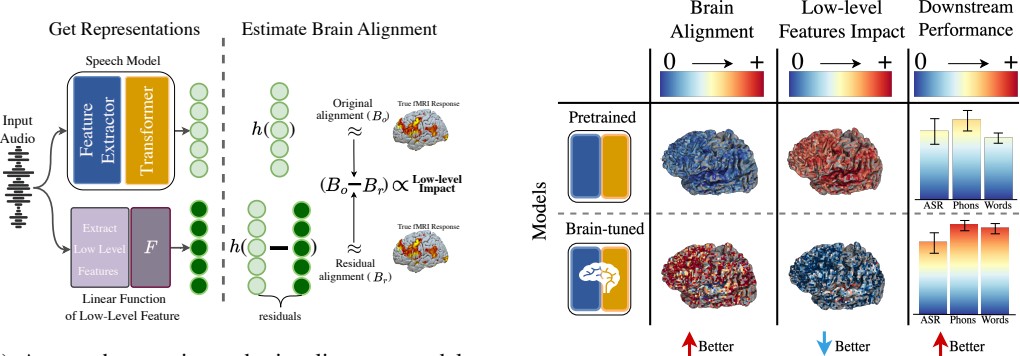

(b) Approach to estimate brain alignment and low-level feature impact

(c) Evaluation strategy and expected outcomes

Figure 1: Training and Evaluation Approaches. (a) Brain-tuning approach for a given speech model; (b) Evaluation of brain alignment and low-level feature impact on the brain alignment; (c) Types of evaluation and expected outcomes if brain-tuning successfully improves semantic understanding in speech models: increase of alignment with semantic brain regions, decrease of impact of low-level features on this alignment, and increase in downstream performance on semantic tasks.

is expected to represent lower-level information than the decoder, it is a good testbed for whether brain-tuning can induce semantic understanding.

## 3.2 NATURALISTIC BRAIN DATASET AND DATA PREPROCESSING

We use the largest public dataset of fMRI recordings (LeBel et al., 2024) for brain-tuning. The dataset contains fMRI recordings for 8 participants listening to 27 short stories from the Moth Radio Hour podcast for a total of 6.4 hours of audio per participant ($11,543$ fMRI images (TRs) with TR = 2.0045s). To fine-tune a model using fMRI recordings, we need to build a paired dataset of fMRI recordings and the corresponding audio snippets that were presented to the participants. We follow previously proposed approaches for this (Oota et al., 2024a; Vaidya et al., 2022; Antonello et al., 2024; Schwartz et al., 2019). Specifically, we first partition the audio input by utilizing a sliding window of length $T$ seconds with a stride $W$ seconds. This way, at each time $t$ in the audio, a window of length $[t - T, t]$ seconds is provided as input to the speech model. We use $T = 16$s and $W = 0.1$s. We next align the stimulus presentation rate with the slower fMRI acquisition rate by downsampling using a 3-lobed Lanczos filter. Lastly, we account for the slowness of the fMRI hemodynamic response by modeling it as a finite response filter with 10 seconds (5 TRs). These steps result in an audio-fMRI paired dataset that can be used for brain-tuning or evaluation.

**Estimated noise ceiling.** Noise in fMRI data can impair brain-tuning and evaluation, so it is important to estimate the "noise ceiling" of each voxel in the fMRI recordings. We estimate the voxel-wise noise ceiling for all participants' fMRI data based on the preferred method by the original dataset paper (LeBel et al., 2023), which leverages within-participant repetitions of the same story. This noise ceiling value estimates the amount of explainable variance in the brain signal, ranging from $0$ to $1$. We use this estimated noise ceiling to filter noisy voxels and to normalize the brain alignment during evaluation. We use a filtration threshold of $0.4$, in line with the findings of Antonello et al. (2024). After filtering voxels with low noise ceiling, there remain $30,000$ to $50,000$ voxels per participant.

The final brain-tuning voxel set contains voxels from late language regions and the auditory cortex. Note that because the late language regions are much larger than the auditory cortex, the number of included voxels from the late language regions is naturally much greater (as shown in Fig.13).

### 3.3 BRAIN-TUNING SPEECH MODELS

**Brain-tuning approach.** Given an input audio and its corresponding fMRI response, obtained via the method in Section 3.2, we aim to fine-tune a pretrained speech model with the fMRI responses (i.e., brain-tune the model). Specifically, we fine-tune the model to reconstruct the fMRI responses corresponding to the voxels with high noise ceiling ($> 0.4$). The approach is illustrated in Fig.1a. To this end, we add a pooling layer and a projection head on top of the output tokens. The projection head predicts the fMRI response from the pooled model tokens. More formally, given the $o_1....o_N$ output tokens, we have a function $\mathbf{H}$, that predicts fMRI targets such that $\mathbf{H}(o_1 : o_N) = FC(P(o_1 : o_N))$, where $P$ is an average pooling function and $FC$ is a linear function. The training objective is a reconstruction loss ($L_2$ loss) between the outputs of $\mathbf{H}$ and the fMRI voxels. We freeze the feature extractor and backpropagate the loss to fine-tune the projection head and the transformer layers.

**Training details.** We used a base learning rate of $5 \times 10^{-5}$ and $10^{-4}$ respectively for the transformer layers and the linear projection head. Both had a linear decay scheduler for the learning rate with a warmup period for $10\%$ of the epochs. The 27 fMRI stories are split into a training set (24 stories), a validation set (2 stories), and a held-out test set (1 story). The training is stopped when the validation loss saturates or begins to diverge. Since the number of voxels differs for each participant, this fine-tuning process is done separately for each fMRI participant. We apply this approach to the 3 pretrained models: Wav2vec2.0, HuBERT, and the Whisper encoder.

#### 3.3.1 COMPARISON MODELS

In addition to comparing the brain-tuned models to the corresponding pretrained ones, we further train several additional baselines for comparison. We briefly summarize these baselines and their purpose below, and provide more details about each baseline in Appendix D.2.

**Random brain-tuned.** This baseline aims to test how the addition of any fMRI data impacts model performance. This baseline uses the same fine-tuning process as in Fig.1a, but instead of using the matched fMRI responses for the input stimulus, it uses block-permuted fMRI responses.

**Big spoken language model-tuned (BigSLM-tuned).** This baseline tests the importance of having fMRI responses as the training targets. We replace the fMRI targets for the input stimuli with representations for the same stimuli obtained from a BigSLM. We use Whisper Medium (800M parameters) as the BigSLM and use a concatenation of all its decoder layers' representations.

**Stimulus-tuned**. This baseline tests whether tuning with the fMRI signal results in additional gains over simply further tuning only using the stimulus audio. Stimulus-tuned models have been previously found to outperform pretrained models specifically for brain alignment (Merlin & Toneva, 2022), but their performance on downstream tasks has not been investigated.

**Text language model-tuned (LM-tuned)**. We expect that current text LMs encode richer semantics than current speech LMs, so this baseline tests the importance of added semantics for model performance. For tuning, we use representations from two pretrained text LMs (GPT2 and LLama2). We leverage LM-tuned models to detect which downstream tasks benefit from more semantics.

In the main paper, we focus on two of these baselines–Random Brain-tuned and BigSLM-tuned–and provide results from the remaining baselines in Appendix D.2. Briefly, stimulus-tuned models perform similarly to pretrained models and substantially worse than brain-tuned models on the tested downstream tasks. LM-tuned models improve over the pretrained models on two downstream tasks, the same ones where brain-tuning leads to the biggest gains over the pretrained models. This further supports our conclusions that brain-tuning improves semantic understanding in speech models.

### 3.4 EVALUATION

We evaluate multiple aspects of the brain-tuned models and illustrate our evaluation strategy in Fig.1c. If brain-tuning successfully improves semantic understanding in speech models, we expect

that brain-tuned models will align better with semantic language regions in new brain recordings, have impact of lower low-level features on the alignment with these regions, and have improved downstream performance on semantic tasks.

### 3.4.1 BRAIN ALIGNMENT

To compare brain alignment for a model before (i.e., the pretrained version) and after brain-tuning, we compute the normalized brain alignment using standard voxel-wise encoding models and report it for language- and speech-related brain regions. For each region, we statistically test whether brain-tuning leads to significantly better alignment.

**Normalized brain alignment.** We estimate standard voxel-wise encoding models to evaluate the brain alignment of a model representation (Antonello et al., 2024; Vaidya et al., 2022; Oota et al., 2024a). We carry out this voxel-wise encoding as shown in the original alignment branch in Fig.1b. The audio data is processed as detailed in Section 3.2, then a voxel-wise encoding function $\mathbf{h}$ is learned using ridge regression on the training portion of the dataset. The prediction performance of this encoding function is computed over the held-out testing portion of the dataset via Pearson correlation. For a voxel $v$, we define $\rho_v$ (the alignment for voxel $v$) as the Pearson correlation between the predictions of $\mathbf{h}$ and the corresponding brain responses for this voxel across all held-out data samples. Lastly, we define the normalized brain alignment $B$ for a brain region of $V$ voxels as:

$$B = \frac{1}{|V|} \sum_{v \in V} \frac{1}{NC_v} \rho_v \tag{1}$$

where $NC_v$ is the noise ceiling for voxel $v$. This serves as a standardized measure for alignment between a model and different brain regions since it is computed relative to the estimated explainable variance in the brain region.

**Parsing language and primary auditory regions.** To make the normalized brain alignment comparison focused on language and primary auditory regions, we use FreeSurfer v7 to project the participants' data, and then we use the human cerebral cortex parcellation atlas from (Glasser et al., 2016) to parse the regions of interest (ROIs). We focus mainly on the late language regions (e.g., inferior frontal gyrus, angular gyrus, anterior and posterior temporal lobes, and middle frontal gyrus) and the primary auditory regions. The full ROI list and their functions is provided in Appendix A.1.

**Significance testing.** To test whether the brain-tuned models have significantly different alignment than the pretrained ones, we use the Wilcoxon signed-rank test. We indicate significant differences (corresponding to p-value $< 0.05$) with an asterisk *.

### 3.4.2 IMPACT OF LOW-LEVEL FEATURES ON BRAIN ALIGNMENT

Previous work showed that the alignment of pretrained speech models with late language regions is mostly due to low-level features (Oota et al., 2024a), which is undesirable. We further set out to test the impact of low-level features on the brain-tuned models' alignment with the brain. To enable comparisons with previous work, we estimate the low-level feature impact on brain alignment using the same approach as in Oota et al. (2024a). Intuitively, the impact of a specific low-level feature is estimated by comparing the brain alignment of a model before and after this low-level feature is computationally removed from the model. If, after removal of the low-level feature, the alignment is significantly lower than the original one, the low-level feature is said to have high impact on the brain alignment. We illustrate this process in Fig.1b and provide details about this method below.

**Low-level features.** We focus on four low-level speech features: Power Spectrum (the time-varying power spectrum across frequency bands), Di-Phones & Tri-Phones (adjacent pairs and triples of phonemes), and Articulation (articulatory characteristics of the phonemes). These features cover different stages of speech and are considered to be non-semantic features. The specifics of obtaining these features from the audio are detailed in (Oota et al., 2024a) and Appendix A.3.

**Low-level feature impact.** First, given a low-level feature of the input audio, a linear function $\mathbf{F}$ learns to predict the representations of the model from this feature. Then, the predicted model representations are subtracted from the true representations, and the brain alignment of this residual is estimated via a standard encoding model (Section 3.4.1). We define the **low-level impact** $R$ as:

$$R = 100 \cdot \frac{B_o - B_r}{B_o} \tag{2}$$

where $B_o$ and $B_r$ correspond to the original and residual brain alignments. $R$ represents the percentage drop in alignment due to the removed low-level feature. Large $R$ means that much of the original alignment was due to the low-level feature. To test for significant differences between models, we perform the same statistical tests as described in Section 3.4.1.

### 3.4.3 DOWNSTREAM TASKS

To test whether improving brain-relevant semantics via brain-tuning also improves semantic understanding in models, we evaluate our models on a range of downstream tasks at different semantic levels. We also test the semantic vs. phonetic preference of the models' representations.

**Downstream tasks.** We choose tasks with several semantic difficulties, namely: automatic speech recognition (ASR), phonetic sentence type prediction, sequence understanding, phonemes prediction, word identity prediction, and emotion recognition. We use standard datasets for these tasks: TIMIT (Garofolo, 1993), Crema-D (Cao et al., 2014), Speech Commands (Warden, 2018), and SLURP (Bastianelli et al., 2020). All datasets were not seen by the model during brain-tuning. Appendix B details further information about the datasets and the formulations for each task. We consider emotion recognition to be the least semantic as tone and prosodic information are highly predictive of emotions in speech (Singh & Gupta, 2023). Phonemes and word prediction are moderate in semantic difficulty, and the rest are high in semantic difficulty as they require understanding beyond single-word/phone decoding. Additionally, to have a more empirical guide on which task we expect a model with improved semantic understanding to perform better on, in Appendix D.2 we provide downstream results from the text LM-tuned baselines. We observe that text LM-tuned models benefit two tasks in particular: phonetic sentence type prediction and phonemes prediction, suggesting that these two tasks can benefit most from semantics.

**Downstream evaluation.** To perform downstream analysis, we add a linear projection head $f$, where the input is the layer representation and the output is task-specific (e.g., which phonemes were present, which word was present, etc). Each task performance is evaluated on held-out data, provided by each dataset, and an aggregate performance metric is reported. Except for ASR, all tasks use linear probes across layers and are evaluated using the F1-score on a held-out test set. Since they are classification tasks, we also report an additional Naive classifier as a comparison baseline that predicts the majority class for the given task. For ASR, we fine-tune the whole transformer model, calculate the Word Error Rate (WER) on the held-out test set, and report $(1 - WER)$ as the performance accuracy metric.

**Semantic-phonetic preference.** Previous work has shown that speech models' representations are consistently more phonetic than semantic across all layers (Choi et al., 2024). They also show that even in a seemingly semantic task such as Intent Classification (IC), the models rely on phonetic not semantic features to do the task. We further test the semantic-phonetic preference of our models, using the same method as Choi et al. (2024), which we detail in Appendix C. Briefly, the method tests the representation distance between a set of words, their phonetic neighbors (e.g. "divine" and "divide"), and their semantic neighbors (e.g. "divine" and "god"). A model for which phonetic neighbors are closer than semantic ones is said to have a phonetic preference.

## 4 RESULTS

### 4.1 BRAIN ALIGNMENT WITH HELDOUT DATA

We estimate the normalized brain alignment described in Section 3.4.1 separately for two important language-related areas of the brain: the late language regions and the primary auditory regions. The late language regions are thought to support semantic language processing, while the primary auditory regions support mostly lower-level processing related to the speech signal (Deniz et al., 2019). For each of the three model families, we evaluate the normalized brain alignment for the pretrained and brain-tuned versions, along with the alignments of two main comparison baselines– BigSLM-tuned and Random Brain-tuned (see Section 3.3.1).

In Fig.2a and 2b, we show the normalized brain alignment averaged across voxels, layers, and participants for all models. We observe that brain-tuning significantly improves alignment with late language regions for the self-supervised models (Wav2vec2.0 and HuBERT), with an increase of

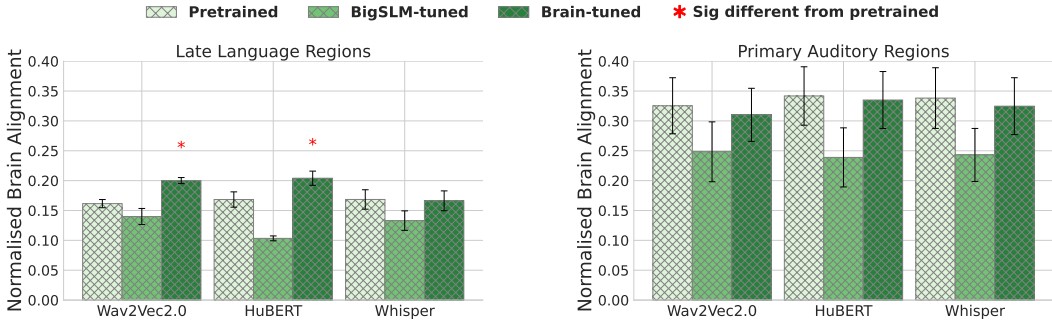

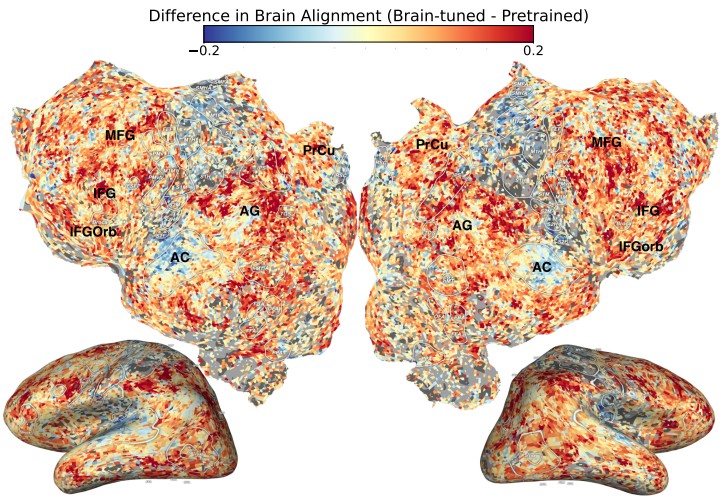

(c) Difference in brain alignment due to brain-tuning of Wav2vec2.0

Figure 2: (a), (b) Mean normalised brain alignment for different brain areas. Error bars indicate the standard error across participants, with * indicating significantly different alignment from pretrained. Brain-tuning significantly improves alignment with late language regions for the self-supervised models. (c) Voxel-wise differences in brain alignment between brain-tuned and pretrained Wav2vec2.0 for a representative participant. Higher alignment is observed in semantic areas.

30% over the corresponding pretrained models. This gain in alignment with late language regions can also be seen on the level of individual voxels (Fig.2c for Wav2vec2.0 and one representative participant; the brain maps for the remaining participants are shown in Appendix F.2). In contrast, the two comparison models–BigSLM-tuned and Random Brain-tuned (see Appendix Fig.6 for Random Brain-tuned results)–lead to lower brain alignment than the pretrained models. This suggests that the gain from the brain-tuned models is due to incorporating the correct fMRI signal that corresponds to the audio input. We do not observe significant gains for Whisper in the late language regions or for any of the model families in the primary auditory regions.

The result that brain-tuning improves the alignment of two of the pretrained models with semantic late language regions, and not with less semantic regions, such as the primary auditory cortices, suggests that brain-tuning may improve the brain-relevant semantics in at least some speech language models. We test this further in the next sections.

## 4.2 EFFECT OF LOW-LEVEL FEATURES ON BRAIN ALIGNMENT

We further test the dependence on low-level features of the observed gain in brain alignment due to brain-tuning. Fig.3a and b present the impact of low-level features on the brain alignment across model families (averaged over voxels, layers, low-level features, and participants).

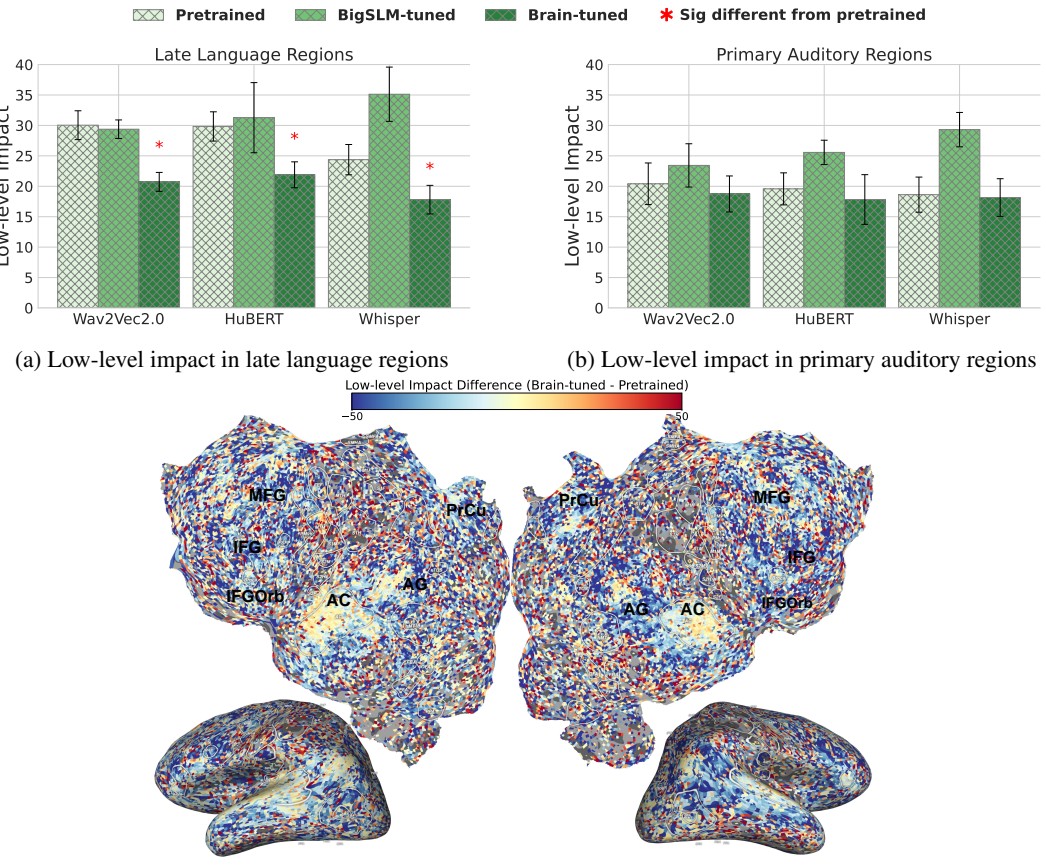

(a) Low-level impact in late language regions
(b) Low-level impact in primary auditory regions

(c) Difference in low-level impact due to brain-tuning of Wav2vec2.0

Figure 3: (a), (b) Mean impact of low-level speech features (percentage drop in brain alignment) for different regions. Error bars indicate the standard error of the mean across participants, and * denotes significantly lower low-level impact than in the pretrained model. All models have significantly lower low-level impact in late language regions. (c) Voxel-wise differences in low-level impact between brain-tuned and pretrained Wav2vec2.0 for a representative participant.

We observe that brain-tuning reduces the impact of low-level features on alignment with late language regions across all model families, including Whisper, which did not originally show improvements in the total brain alignment in these regions (Fig.2a). The comparison models once again do not account for the improvements due to brain-tuning. Similarly to before, for all models, brain-tuning leads to no significant changes related to the primary auditory regions. These observations also hold on the voxel-level (Fig.3c for Wav2Vec2.0 and one representative participant; see Appendix F.3 for the remaining participants). Moreover, brain-tuning can also make up for scale: we observe that a brain-tuned smaller model (HuBERT small) is similarly aligned with late language regions as a substantially larger pretrained model of the same family (HuBERT large), while having lower impact of low-level features on this alignment (see Appendix A.2 for more details).

Overall, these results indicate that brain-tuning's gain in alignment with the late language areas is not merely in the magnitude but also in its nature, as the reliance on low-level features for alignment with semantic regions is reduced. Next, we investigate if the induced brain-relevant semantics via brain-tuning also lead to improvement in semantic understanding of the models.

## 4.3 DOWNSTREAM PERFORMANCE

We next investigate the models' semantic understanding via their performance on downstream tasks (detailed in Section 3.4.3), and their semantic-phonetic preference (detailed in Appendix C).

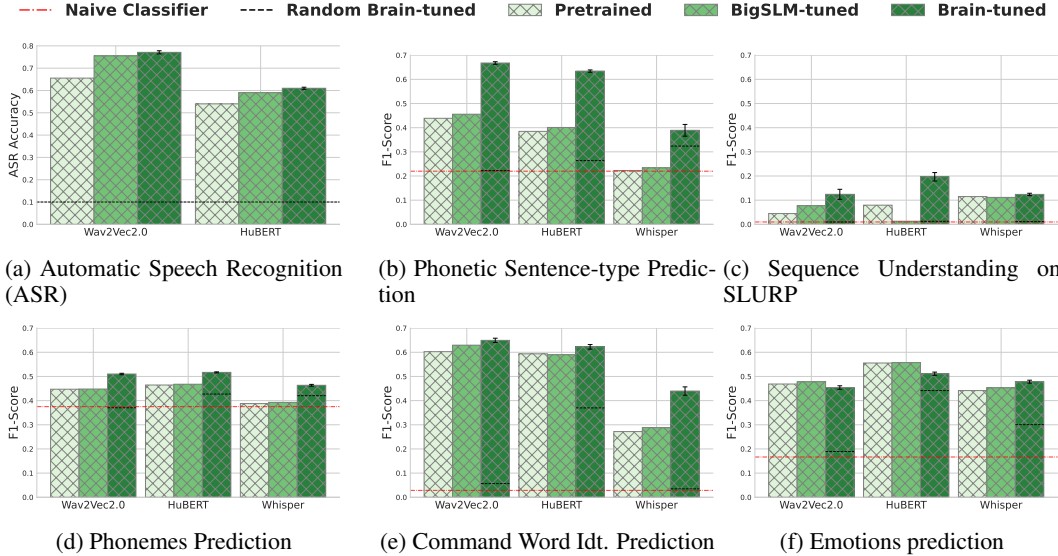

Figure 4: Downstream task performance for different models. Brain-tuned models' performance is the mean and STE across participants. Brain-tuned models show consistent improvement over the baselines, with the biggest gains in more semantic tasks.

**Performance on downstream tasks.** Fig.4 presents the models' performances on popular downstream tasks that require different levels of semantic understanding. The brain-tuned performance is averaged across layers and participants, except for ASR where the performance is only averaged across participants since evaluations are done only once per model. We observe that brain-tuning boosts the performance of the pretrained models for all model families across all tested downstream tasks, with the exception of emotion prediction, in which the performance improves slightly for one model family and decreases slightly for the other two (Fig.4f). The brain-tuned models also generally outperform the BigSLM-tuned baselines, with the exception of the emotion prediction task. Most notably, brain-tuning leads to the most consistent gains above the BigSLM-tuned and pretrained baselines for those tasks where fine-tuning with representations from text-based language models is also helpful (Fig.10; Appendix D.2). This provides additional evidence that brain-tuning is most beneficial for tasks where improved semantic understanding is helpful.

**General trends in downstream performance.** We observe a few trends in all downstream tasks: (1) the BigSLM-tuned models perform on par or better than the pretrained ones, indicating that the BigSLM-tuned model is a meaningful baseline. (2) Generally, the pretrained, brain-tuned, and BigSLM-tuned models perform better than the Random Brain-tuned baselines, which indicates that permuting fMRI targets does not lead to improvement on downstream tasks. (3) The models perform better than naive classifiers, so substantial gains in performance cannot be attributed to random behavior. (4) More interestingly, the brain-tuned Whisper encoder consistently and substantially outperforms its pretrained and BigSLM-tuned versions, which often have close to random or naive performances. This shows that the substantial gain in performance is due to fine-tuning with the matched fMRI data.

**Change in semantic-phonetic preference.** Finally, when we compare the semantic-phonetic preference for the brain-tuned vs. pretrained models (Fig.8), we find that the preference becomes more semantic in the late layers for the brain-tuned models. In contrast, the pretrained models show either no change or a decrease in semantic preferences. This indicates that the brain-tuned models have reduced phonetic preference in the later layers. We elaborate on these findings in Appendix C.

Excitingly, the consistent improvement in downstream semantic tasks and reduced phonetic preference in late layers is in tandem with the improvement in brain alignment with late language regions and the reduced impact of low-level features on this brain alignment. These results provide converging evidence for improved semantics in the brain-tuned models.

## 5 DISCUSSION

In this work, we present a method to augment speech model training directly with fMRI recordings of people listening to natural stories and show two converging lines of evidence that this leads to improved semantic understanding in the models.

First, two of the three tested model families have improved alignment with new brain recordings in the semantic language regions after brain-tuning (Fig.2). For all model families, brain-tuning also significantly reduces the impact of low-level speech features on alignment with late language regions (Fig.3). This is a marked improvement over pretrained models, which were shown to almost entirely rely on low-level speech features to align with semantic brain regions (Oota et al., 2024a). Furthermore, brain-tuned models align with semantic language regions as well as larger pretrained models while also having a lower low-level feature impact (Appendix Fig.5). This suggests that brain-tuning has the potential to break the brain alignment plateau for speech models, which has been shown to reach saturation around 700M parameters (Antonello et al., 2024). Brain-tuning larger models may break this saturation and lead to substantially more alignment in the semantic language regions, further increasing their utility as model organisms (Toneva, 2021).

Second, the brain-tuned models most substantially improve performance on downstream semantic tasks while also maintaining performance on core speech low-level tasks (Fig.4 & Fig.9). This provides additional evidence that brain-tuning improves semantic understanding in speech models while not negatively impacting their fundamental speech capabilities. We attribute the brain-tuned models' balanced semantic and low-level understanding to the inclusion of voxels from both late language areas and auditory cortex during brain-tuning. We expect that if brain-tuning only considered auditory cortex voxels, there would not be improvements in the model's semantic understanding. Conversely, if brain-tuning only considered late language areas voxels, the model's speech performance would substantially degrade. Future work can investigate these hypotheses.

Lastly, the brain-tuned models also improve their semantic preference in the late layers (Appendix Fig.8). We further emphasize that the gain in performance and semantic preference is due to additional training data of less than $0.7\%$ of the original training size, which is comparable to, if not less than, a typical fine-tuning dataset for a specific task. Yet, in the brain-tuning case, there is an evident gain in performance and a wider generalization across multiple downstream semantic tasks. There is also good evidence that expanding brain data and model families further will increase the gains from brain-tuning (Appendix E.2 & E.1). Overall, to the extent of our knowledge, this is the first work to show substantial downstream performance gains when augmenting an ML model with brain-relevant data, across not only the speech domain but also language and vision.

Both lines of evidence show that brain-tuning not only increases the raw performance metrics but also changes the representations of the model to rely less on low-level features when performing a task that relies on semantics. These findings also strengthen the utility of brain alignment in determining how semantically oriented a given model representation is. Lastly, all results signal that there is still room for improvement in the speech models' semantic capabilities, and we hope that our work inspires future work that improves on our brain-tuning method.

## 6 CONCLUSION

Our systematic analyses of the utility of brain-tuning on semantic brain alignment and downstream performance reveal a parallel among the gain in brain alignment, its reduced impact of low-level speech features, and increased downstream performance on several tasks with varying semantic difficulty. We further observe an increase in the semantic preference of late layers of the brain-tuned models. These exciting results provide evidence for the first time that incorporating brain signals into the training of language models improves their semantic understanding. Future work can investigate further refinement of the brain-tuning loss and the incorporation of additional participants and brain datasets in the brain-tuning process. We hope that the brain-tuned models we provide will serve as better model organisms for auditory language processing in the brain, and will inspire more work on improving the alignment between language in machines and language in the brain.

## REPRODUCIBILITY STATEMENT

In this paper, we fully describe our proposed brain-tuning approach and how to evaluate it. (1) Sections 3.1, 3.2, and 3.3 detail the exact model families, data processing, and fine-tuning settings and hyper-parameters needed to carry out Brain-tuning for any given model. (2) Section 3.4, Appendix B, Appendix C, and Appendix D.2 detail all evaluation pipelines and metrics, alongside any additional datasets or training settings and hyper-parameters needed for evaluation. (3) We will make the code for brain-tuning and its evaluation publicly available once the paper is accepted.

## ACKNOWLEDGMENTS

This work was partially funded by the German Research Foundation (DFG) - DFG Research Unit FOR 5368 and by the CS@max planck graduate center.

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

## A    BRAIN ALIGNMENT

### A.1    ROIs DETAILS

The human cerebral cortex multi-modal parcellation (Glasser Atlas) has 180 labeled ROIs per hemisphere(Glasser et al., 2016). It has language regions that include the following labels: Angular gyrus (AG: PFm, PGs, PGi, TPOJ2, TPOJ3), lateral temporal cortex (LTC: STSda, STSva, STGa, TE1a, TE2a, TGv, TGd, A5, STSdp, STSvp, PSL, STV, TPOJ1), inferior frontal gyrus (IFG: 44, 45, IFJa, IFSp) and middle frontal gyrus (MFG: 55b) ((Oota et al., 2024a), Desai et al. (2023)). It also has the primary auditory (A1) and the early auditory (A1, PBelt, MBelt, LBelt, RI, A4) regions.

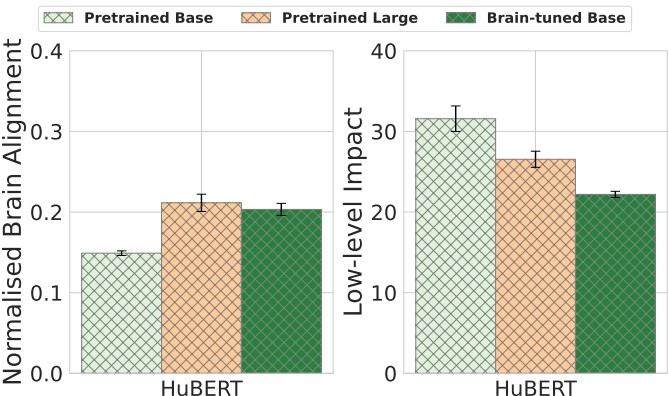

Figure 5: Brain alignment and low-level impact comparison with pretrained HuBERT large architecture on 3 subjects. The HuBERT base architecture performs closely to the large pretrained architecture and is less affected by removal of low-level features

### A.2    BRAIN-TUNING ALIGNMENT COMPARISON WITH BIGGER MODELS

To get a clearer idea about the improvement in brain alignment we get from Brain-tuning on late language regions, we compare the HuBERT model's base architectures ( 90M parameters ) both pretrained and brain-tuned to the pretrained HuBERT large architecture (320M parameters) in Fig.5. The pretrained large architecture has a much larger alignment in late language regions than the pretrained base model, which is in line with the trend of increase in alignment when scaling the model size shown in (Chung et al., 2024); however, the Brain-tuned base model is very close to the pretrained large model's alignment. Moreover, the low-level impact of the Brain-tuned model is noticeably less than the pretrained large model. This is an indicator that we can explain much more in the brain late language semantic regions with smaller models and brain-tuning bigger models might break the plateau we see in (Antonello et al., 2024) when reaching huge model sizes. Breaking this plateau might allow for better and more accurate computational models for speech semantic processing in the brain.

### A.3    LOW-LEVEL SPEECH FEATURES DETAILS

Here, we detail the definition and acquisition method of the low-level speech features used in our computation of low-level impact (namely Power Spectrum, Di-Phones & Tri-Phones, and Articulation ). We obtain these features for the fMRI stories used for brain alignment computation (Section 3.4.1).

**Power Spectrum.** We follow the method described in (Gong et al., 2023). For each TR time (a 2s segment), we quantify the time-varying power spectrum across 448 frequency bands. This power spectrum is obtained by estimating the power of the sound signal between 25 Hz and 15 kHz, in 33.5 Hz bands (giving a total of 448 bands). For our experiments, we use the power spectrum features from (Deniz et al., 2019).

**Di-Phones & Tri-Phones.** For a given speech utterance, DiPhones represent the adjacent pair of phones (e.g., [da], [aI]). For each audio segment of length 2 seconds (TR length), we have a one-hot encoding vector representing the presence or absence of all possible 858 diphones. The same is done for Tri-Phones but for triplets of phones instead of pairs. The annotations of this data were done using the Praat software and were obtained from the shared data by the authors of (Oota et al., 2024a).[1]

**Articulation.** Phoneme articulations are the specific ways speech organs (e.g., tongue or vocal cord) move to produce different phonemes. They associate each phoneme with a set of properties that specify things like whether this phoneme makes vocal cords vibrate. We use phoneme articulations as mid-level speech features, mapping hand-labeled phonemes to a set of 22 articulatory characteristics. For our experiments, we use the articulation annotations from (Deniz et al., 2019).

## A.4 BRAIN ALIGNMENT WITH RANDOM BRAIN-TUNED

We extend Fig.6 by showing the Random Brain-tuned baseline alongside the brain-tuned, pretrained, and BigSLM-tuned models. Fig.6b clearly shows that Random Brain-tuned models have the lowest brain alignment values in late language regions compared to even BigSLM-tuned baselines, and in the primary auditory regions, it's also much lower than the pretrained version. This strongly indicates that randomly permuting the fMRI targets while brain-tuning the models with the same stimuli substantially harms the model's alignment. Hence, having the correct fMRI targets is essential for the alignment results we get from the brain-tuned models.

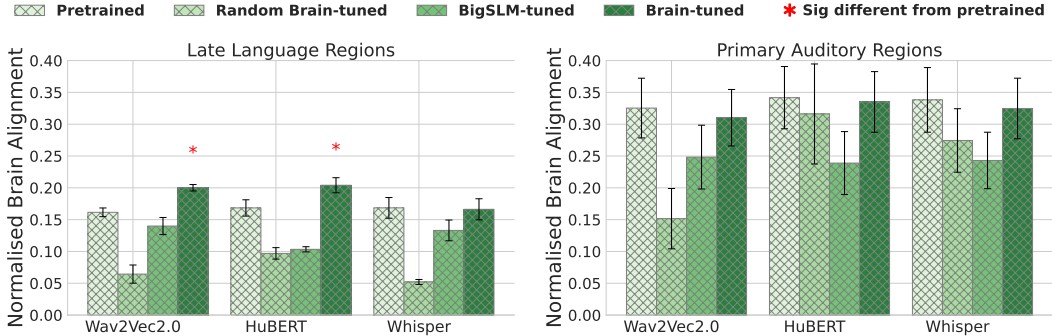

(a) Normalised alignment for Late Language Regions    (b) Normalised alignment for Primary Auditory

Figure 6: (a), (b) Normalized brain alignment across participants and models for different ROIs, where * means brain-tuned has alignment values that are statistically significantly different from pretrained. Random Brain-tuned baselines have much lower brain alignment than pretrained models in the late language and primary auditory areas.

## A.5 LOW-LEVEL IMPACT WITH RANDOM BRAIN-TUNED

We extend Fig.3 by showing the Random Brain-tuned baseline alongside the brain-tuned, pretrained, and BigSLM-tuned models. Fig.7 clearly shows that Random Brain-tuned models have the highest low-level impact (highest drop) compared to even BigSLM-tuned baselines in both late language regions and the primary auditory region. Even though the original normalized alignment values for the Random Brain-tuned models are very low to begin with, they still undergo a substantial drop after we remove the low-level features. Similar to the results from Fig.6, these results also indicate that randomly permuting the fMRI targets while brain-tuning the models with the same stimuli substantially harms the model's semantics. Hence, having the correct fMRI targets is essential for reduced low-level impact we get in the brain-tuned models.

---

[1]https://www.fon.hum.uva.nl/praat/

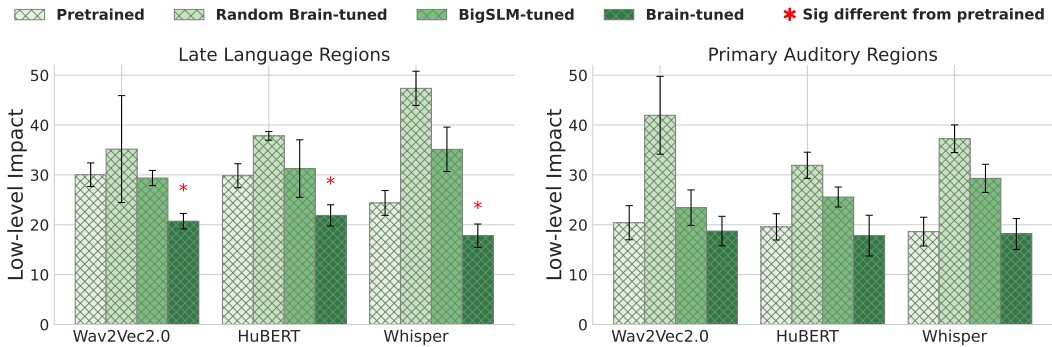

(a) Low-level Impact in Late Language Regions  (b) Low-level Impact in the Primary Auditory Region

Figure 7: (a), (b) Low-level Speech Features Impact (Percentage Drop in Brain Alignment) for different ROIs, where * means brain-tuned has alignment values that are statistically significantly different from pretrained. RandomBrain-tuned baselines have a much higher low-level impact than pretrained models in that late language and primary auditory areas.

## B  DOWNSTREAM TASKS

We detail here the description and dataset used for each downstream task mentioned in Section 3.4.3.

**Automatic Speech Recognition (ASR).**  We run an ASR fine-tuning pipeline for the self-supervised models (Wav2vec2.0 and HuBERT). For ASR we fine-tune the whole model to get the best possible performance. The rationale is to essentially see if the brain-tuning process will make it harder or require longer training to fine-tune the speech model for tasks like ASR. We use a version of TIMIT (Garofolo, 1993) for ASR to carry out this experiment. We use the CTC loss (Baevski et al., 2020) for both self-supervised models and calculate Word-Error-Rate (WER) as the evaluation metric (on the padded test-set). We then report the ASR accuracy as $1 - WER$.

**Phonetic Sentence Type Prediction.**  The TIMIT dataset Garofolo (1993) has 3 different sentence types: SA (Dialect), SX (Compact), and SI (Diverse). The SA sentences are supposed to expose the dialectal variations of the speakers ( and are designed to span all the phonemes of American English); the SX sentences should provide good coverage of phones (phonetically balanced and cover a wide range of phonetic contexts with a small number of words). The SI sentences are anything else (phonetically diverse and more naturalistic). We add a classification head and we use the F1-score for evaluation.

**Sequence Understanding.**  This is very similar to Intent Classification tasks; it tests the ability to understand a sequence. We use the SLURP (Bastianelli et al., 2020) dataset that has audio paired with actions. For example, if the input is "Wake me up at eight o'clock", the action should be "set_alarm". We have 46 possible actions and we add a linear head to predict the action and evaluate using the F1-score.

**Phonemes Prediction.**  Phoneme prediction is formulated as a multi-label classification problem, where the classifier predicts which of the 39 Phonemes were present in the input audio clip. We use the TIMIT dataset Garofolo (1993) for its phonetically rich sentences; we evaluate the test set using the F1-score.

**Word Identity Prediction.**  We want to test the model's ability to decode words from input audio. To simplify this task to befit a classification head, we convert it to a classification task on the Speech Commands dataset (Warden, 2018) which has audio clips, each of which has only one word belonging to a set of 35 commands. The classifier predicts which of the 35 commands were said, and the evaluation is done using the F1-score.

**Emotion Recognition.** We add a classification head for emotion recognition on the CREMA-D dataset Cao et al. (2014); CREMA-D has 7.4K clips from 91 actors, and six different emotions (Anger, Disgust, Fear, Happy, Neutral, and Sad). The classifier predicts which of the 6 emotions is present and is evaluated using the F1-score.

## C    MEASURING SEMANTIC-PHONETIC PREFERENCE

To test if there are semantic changes to the models' representations at a more fundamental level, we try to quantify how a model prefers phonetic over semantic features and compare our brain-tuned models to the pretrained ones. To be able to quantify this difference, we take inspiration from the work by Choi et al. (2024), which found that speech models have a huge bias towards phonetic features. This was done by constructing phonetically similar pairs and semantically similar pairs and then computing their similarity/ distance in the embedding space. Doing this clearly shows that phonetically similar words are closer than semantically similar ones in the embedding space (i.e., the model has strong phonetic preference ). This behavior persists across layers and across models. One aspect of desirable change in that behavior is to have the differences between phonetically and semantically similar pairs lower in the more semantic layers (both should still be better than random), or at least to have a clear hierarchy of that change. To do a similar but more curated analysis, we build a dataset of 2K words, each of word is paired with several semantically similar (e.g., Synonyms) and phonetically similar words (e.g., Homophones). Then, we compute the representational distance between them, namely semantic distance for the distance of the word to the synonym and phonetic distance for the distance of the word to the homophone. After that, we are able to compute a **semantic-phonetic preference** $d$ for any given layer or model. We define $d$ as the average difference between semantic and phonetic distances. Essentially, since we know phonetic distances are smaller from the work by Choi et al. (2024), then when $d$ decreases it means that the gap between semantic and phonetic decreases (semantic is closer to phonetic) and vice versa. If $d < 0$, it means that the given layer is more semantic than it's phonetic. Thus, comparing the values of $d$ for the brain-tuned models and the pretrained ones will tell us if there is a difference in the phonetic preference across layers between them.

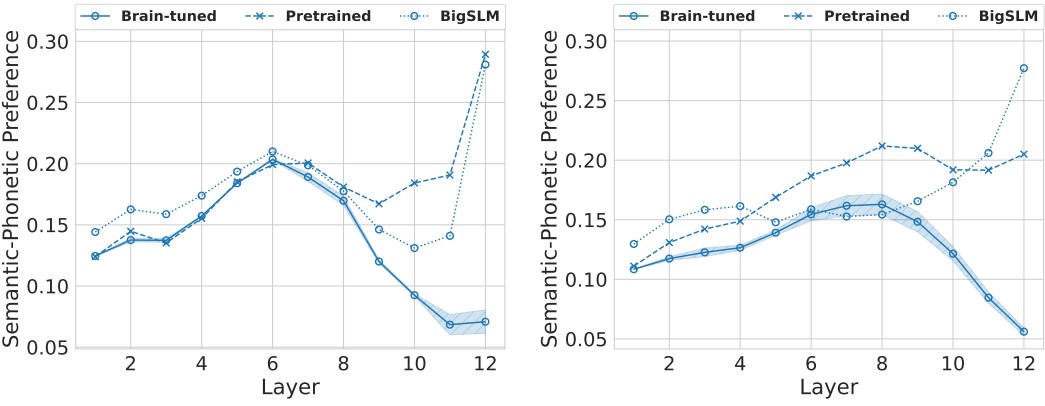

(a) Semantic-phonetic preference for Wav2Vec2.0          (b) Semantic-phonetic preference for HuBERT

Figure 8: Semantic-phonetic preference ($d$) Comparison between brain-tuned and Pretrained models. $d$ decreases in late layers for brain-tuned models, suggesting that the phonetic preference for these layers is decreased.

Finally, when we investigate the semantic-phonetic preference $d$ (the delta of the distances between semantic and phonetic features) in the brain-tuned vs. pretrained models (Fig.8), we find that $d$ decreases in the late layers for the brain-tuned models, but it either increases or stays the same for the pretrained ones. Our findings about the pretrained model replicate those of (Choi et al., 2024). The findings about the brain-tuned models indicate that, unlike the pretrained models, semantic pairs in the late layers of these models are closer to the phonetic ones, and hence the layers are less phonetically dominated (i.e., the semantic preference increases and the phonetic vs semantic preference decreases in these layers). This, together with the downstream results from Fig.4 indicates

that the brain-tuned models are not phonetically biased (in late layers) and that they are capable of performing better on both phonetic and semantic tasks.

# D  ADDITIONAL TASKS AND BASELINES

In this section, we extend the evaluation of brain-tuned models to include two low-level tasks, namely MFCC and FBANK prediction. We also add language model baseline (comparison model) and report its downstream and low-level performance relative to to the other models (i.e., brain-tuned, Pretrained and BigSLM-tuned).

## D.1  BRAIN-TUNED LOW-LEVEL PERFORMANCE

For MFCC prediction, we train linear probes to predict the MFCC coefficients, while for FBANK prediction, we train linear probes to predict the filter banks. For both tasks, we use the TIMIT dataset (Garofolo, 1993) and we evaluate using the $R^2$ coefficient, averaged across layers.

Fig.9 reports pretrained, BigSLM-tuned, and brain-tuned models' performance on two low-level tasks (MFCC prediction and FBANK prediction); it shows that brain-tuned and BigSLM-tuned models don't perform substantially lower than their pretrained counterparts. This indicates that fine-tuned models are still capable of doing these core low-level tasks just as well as their pretrained version. We think this evidence reduces the possibility that catastrophic forgetting happened after fine-tuning.

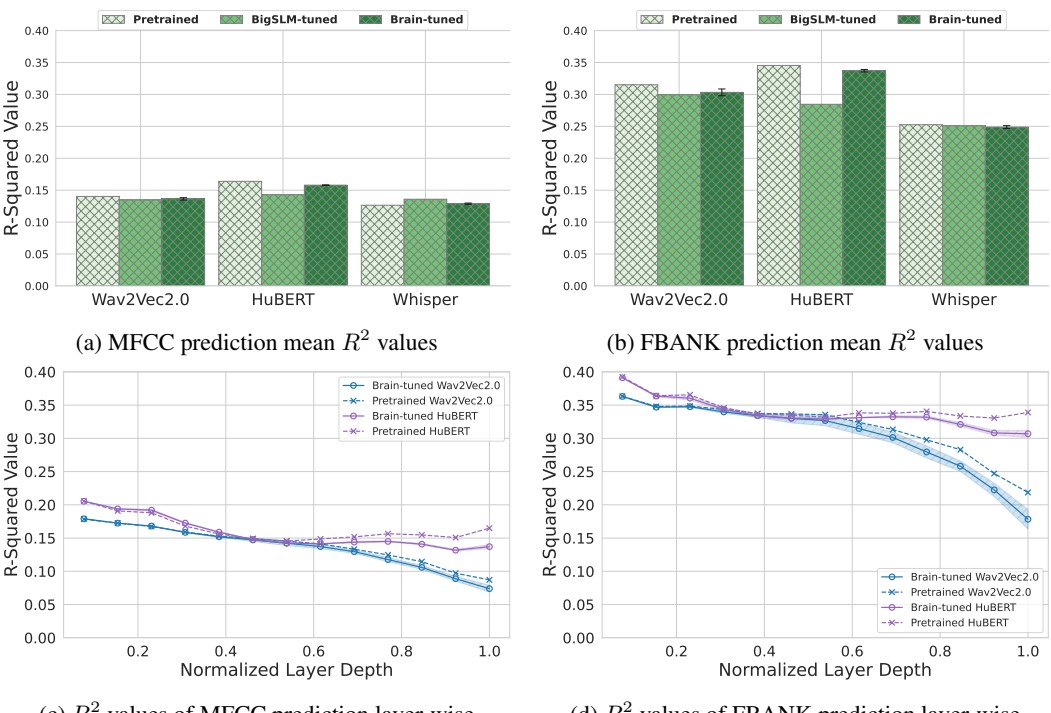

(a) MFCC prediction mean $R^2$ values

(b) FBANK prediction mean $R^2$ values

(c) $R^2$ values of MFCC prediction layer-wise

(d) $R^2$ values of FBANK prediction layer-wise

Figure 9: Low-level task performance comparison for different models. It signifies that brain-tuned models didn't lose low-level capabilities at the expense of gaining more semantics, as the averaged $R^2$ values are very close and the layer-wise values are the same in early layers and slightly decreased in late layers.

### D.2 Details for Comparison Baselines

We elaborate here the comparison models introduced in Section 3.3.1 and provide a complete contrast for all comparison models on all downstream tasks for the Wav2Vec2.0 model family.

**Random brain-tuned.** In order to test how the addition of any fMRI data impacts model performance, this baseline uses the same fine-tuning process for brain-tuning (Section 3.3) but with incorrect randomly chosen fMRI targets. Instead of using the correct fMRI responses for the input stimulus, we use block-permuted fMRI responses that correspond to different stimuli. We carry out the fine-tuning process for each participant, and the training setup is identical to the brain-tuning ones (i.e., the same learning rates, learning rate schedule, optimizer, and number of epochs). The same set of voxels used in brain-tuning is also used for this baseline.

**Big spoken language model-tuned (BigSLM-tuned).** We aim to test the importance of having fMRI responses as the fine-tuning targets with this baseline. To this end, we replace the fMRI target vectors for the input stimuli with representations for the same stimuli obtained from the BigSLM layers. This creates a proxy for the fMRI targets that is rich in meaningful spoken-language related information. For this baseline, we utilize Whisper Medium (800M parameters) as the BigSLM and use a concatenation of all its decoder layers' representations. The fine-tuning objective is then to construct the concatenated representations vectors from the BigSLM layers (instead of reconstructing the fMRI responses). We use the same training setup and the same stimulus data that were used for brain-tuning (Section 3.3).

**Stimulus-tuned.** This baseline tests whether tuning with the fMRI signal results in additional gains over simply further tuning using the stimulus audio only. This highlights any improvements in the model performance that would be only due to training with more data. To isolate the effect of the audio stimulus alone, we develop a model whose pre-training includes the audio stimulus used in the fMRI data (LeBel et al., 2024). To this end, for additional pre-training on the stimulus audio data, we use the same losses with the same weights of Wav2Vec2.0 self-supervised pre-training (the diversity loss and the contrastive loss) (Baevski et al., 2020). This stimulus-tuned model is trained for 200 epochs and uses a base learning rate of $2 \times 10^{-5}$ with a warm-up for the first 10% of the updates and then a linear decay schedule.

**Text language model-tuned (LM-tuned).** We develop two text language model-tuned baselines to test the potential of using a text language model (LM) in a similar fashion to the BigSLM-tuned baseline. The first one uses representations from the pretrained GPT2-Medium model (Radford et al., 2019) which is similar in size to the BigSLM model, and the second one uses representations from Llama2-7B Parameter Model (Touvron et al., 2023) which is a much bigger and more recent large LM. We apply a similar fine-tuning pipeline of brain-tuning with the concatenated representations of the corresponding LM as targets. For the GPT2-Medium model, we concatenate the representations of all layers, while for LLama2-7B one in every 3 layers is taken. We found that LM-tuned models need more training than the BigSLM-tuned one; we train them for 200 epochs. We report the downstream performance of these baselines (the GPT2-tuned and the Llama2-tuned) for the Wav2Vec2.0 model family.

The stimulus-tuned model performs very similarly to the pretrained model, indicating that further self-supervised training on these additional few hours of stimulus data doesn't impact downstream model performance (Fig.10). As for the LM-tuned baselines, GPT2-tuned and Llama2-tuned models improve over the pretrained and BigSLM-tuned ones on two tasks (namely Phonemes and Phonetic Sentence Type Prediction), with the Llama2-tuned being slightly better than the brain-tuned one on the same two tasks. The jump from pretrained to GPT2-tuned to Llama2-tuned indicates that more semantics lead to better performance on Phonemes and Phonetic Sentence Type Prediction tasks. These tasks are also the ones for which brain-tuning leads to the largest improvement over the pretrained models, further supporting our conclusions that brain-tuning can improve semantic understanding in speech models. The results for the Random Brain-tuned and BigSLM-tuned baselines are identical to Fig.4; they mainly emphasize that Random Brain-tuning hurts the model performance and BigSLM-tuning doesn't help strongly with the more semantic tasks.

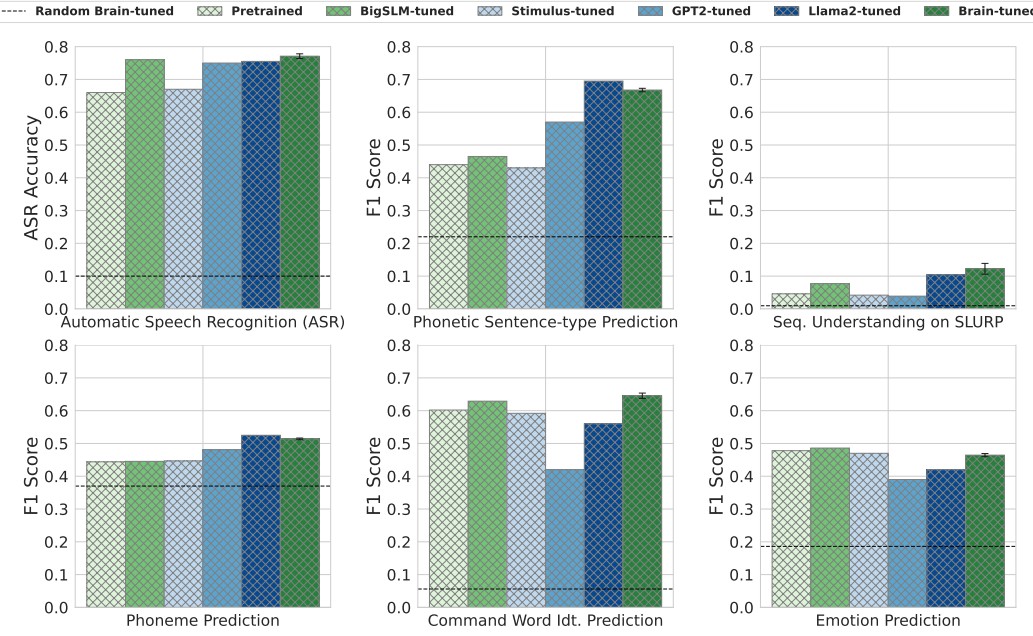

Figure 10: Downstream task performance of brain-tuned models vs. all baselines for Wav2Vec2.0. The additional baselines beyond those presented in Fig.4 are shown in blue hues. We observe that LM-tuned baselines (GPT2-tuned and Llama2-tuned) improve over the pretrained model on tasks where we see big improvements with brain-tuning, further supporting our conclusions that brain-tuning can improve semantic understanding in speech models. The stimulus-tuned model performs very similarly to the pretrained one, indicating that simply further training on the audio stimulus is not sufficient for the gains observed from brain-tuning.

# E    MORE STUDIES ON DOWNSTREAM TASKS

## E.1    DOWNSTREAM PERFORMANCE COMPARISON WITH HUBERT LARGE

To complement our results on the brain alignment gap between HuBERT Large and the Brain-tuned Base version (Appendix A.2), we report here the gap between them in downstream performance (reporting the same participants for brain-tuned models). Fig.11 shows that for HuBERT, the brain-tuned base architecture is closing on the pretrained large architecture. In other words, we gain a performance close to the large architecture with the same number of parameters of the base architecture (after it's brain-tuned).

## E.2    EFFECT OF MODEL AND DATA SIZES ON BRAIN-TUNING

We study here how brain-tuning is affected when we increase the pretrained model size or decrease the size of the data used for brain-tuning. We do this by plotting the performance improvement with brain-tuning (brain-tuning performance − pretrained performance) across model sizes or data sizes, for a range of downstream tasks. Fig.12 shows that as the model size increases, the gain in performance due to brain-tuning on downstream tasks drops. When we lower the amount of data used for brain-tuning, the gain in performance due to brain-tuning also drops. This provides a clear signal that data size plays a key role in the positive results we see from brain-tuning.

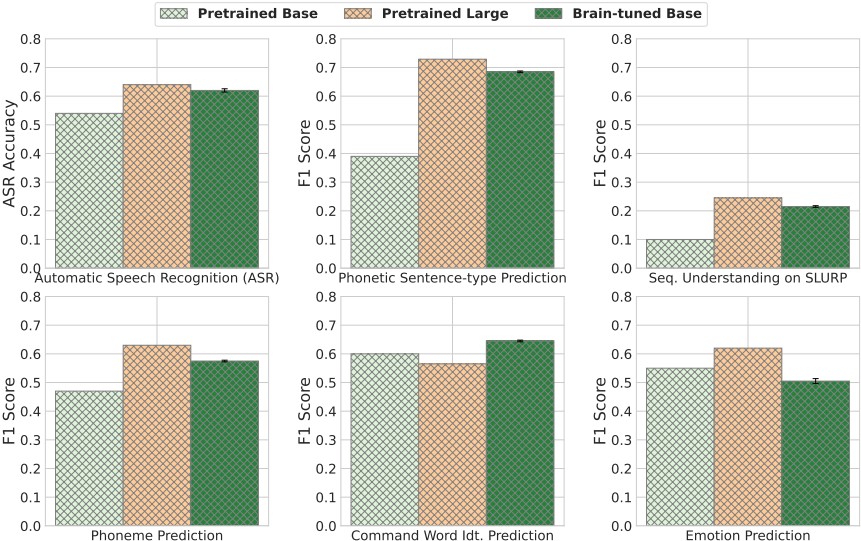

Figure 11: Downstream performance comparison of pretrained HuBERT large architecture vs Brain-tuned HuBERT base architecture. The HuBERT base architecture performs closely to the large pretrained architecture, bridging the gap between the base and large model sizes.

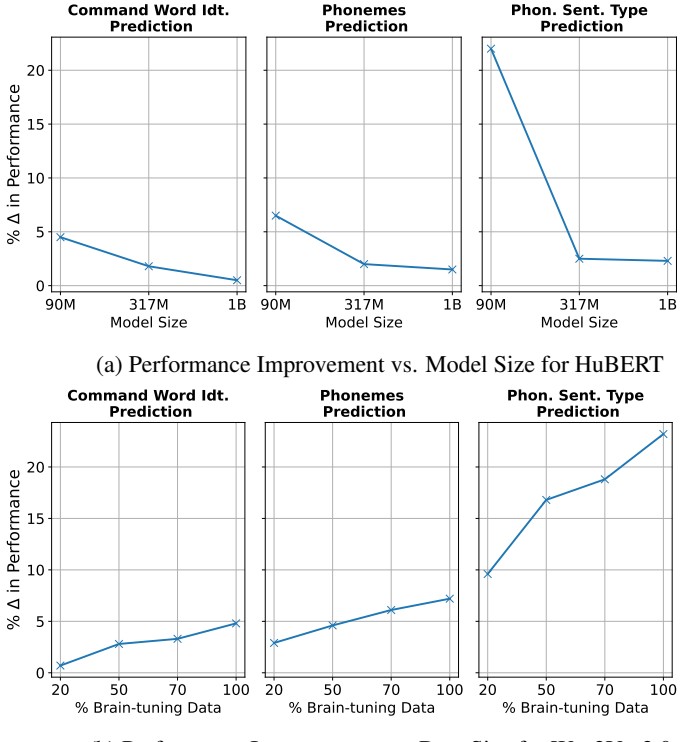

Figure 12: Performance Improvement (brain-tuned − pretrained performance) across model sizes and data sizes. We see clearly that lowering the amount of data limits the gain in performance due to brain-tuning, and the same goes for increasing the model size while keeping the data fixed.

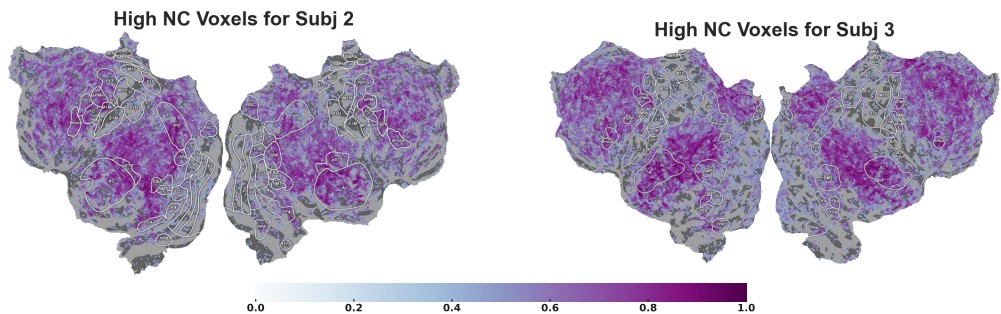

Figure 13: Voxels with Noise Ceiling values above the threshold for Participants 2 and 3. These are the voxels that will be used for brain-tuning

## F    ADDITIONAL BRAIN PLOTS FOR DIFFERENT PARTICIPANTS

Here, we show samples of noise-filtered voxels (the ones that will be used for brain tuning). More-over, we extend the whole-brain plots (flat and lateral views) for different participants for both brain alignment and low-level impact analyses.

### F.1    HIGH NC FILTERED VOXELS

Fig.13 shows the remaining voxels after applying the noise threshold mentioned in Section 3.2 for two participants. These are the voxels that will be used during brain-tuning. Looking at their locations, we see that they cover a large number of semantic regions, as well as the auditory cortex.

### F.2    BRAIN ALIGNMENT VOXEL-WISE DIFFERENCES

We show in Fig.14 more whole-brain analyses for the voxel-wise differences in brain alignment be-tween brain-tuned and pretrained models for different participants. The trend we detailed in section 4.1 is also consistent with the plots below and is in line with the increase in the values of alignment in late language regions (Fig.2a) and the insignificant change in alignment in primary auditory regions (Fig.2b). All shown Brain plots are for the Wav2vec2.0 model family.

### F.3    LOW-LEVEL IMPACT VOXEL-WISE DIFFERENCES

In Fig.15, we show more whole-brain analyses for the voxel-wise differences in low-level impact due to brain-tuning (between brain-tuned and pretrained models) for different participants. The trend we detailed in section 4.2 is also generally consistent with the plots below and is in line with the increase in the values of alignment in late language regions (Fig.3a) and the insignificant change in alignment in primary auditory regions (Fig.3b). All shown Brain plots are for the Wav2vec2.0 model family.

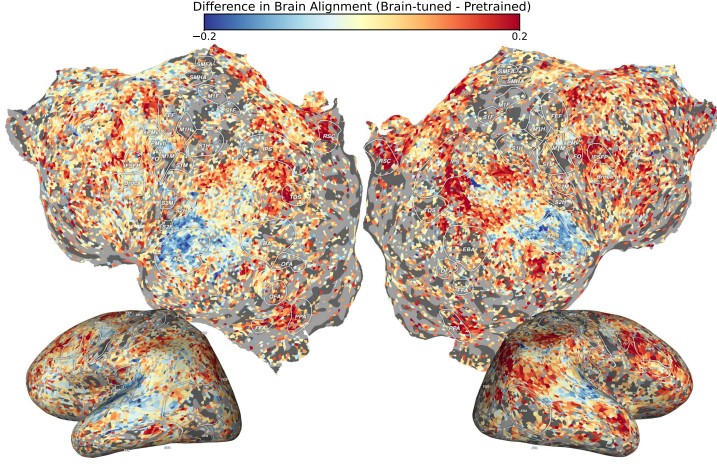

(a) Difference in Brain Alignment for Participant 1

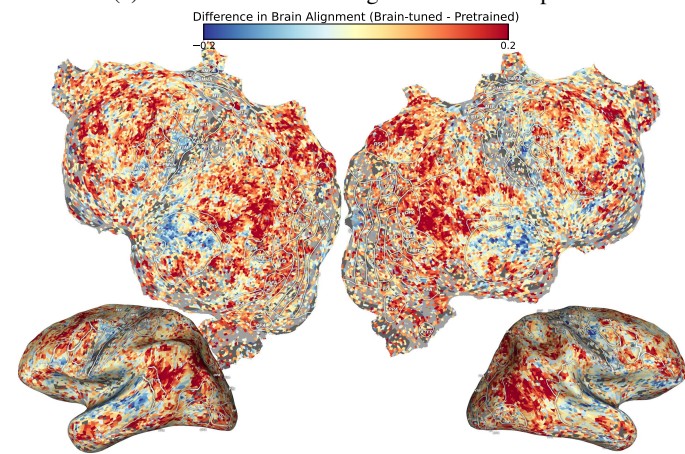

(b) Difference in Brain Alignment for Participant 2

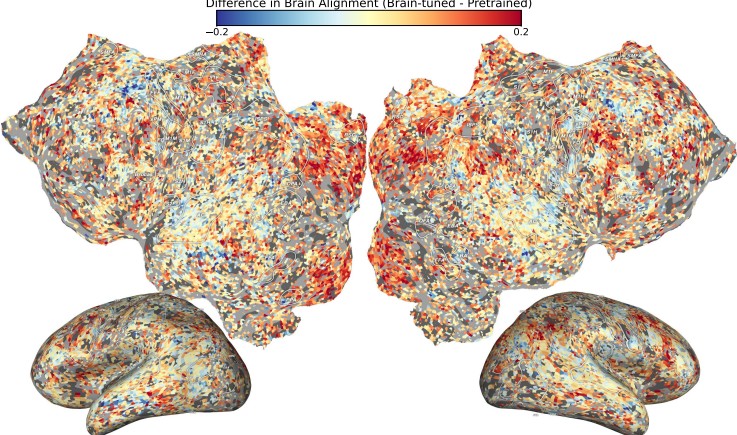

(c) Difference in Brain Alignment for Participant 6

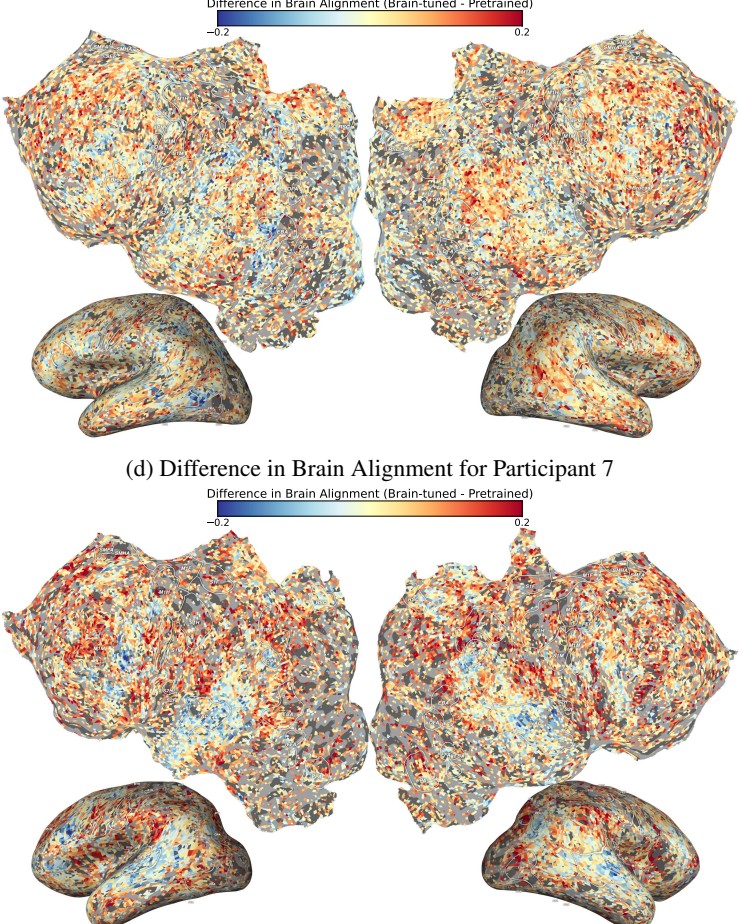

(d) Difference in Brain Alignment for Participant 7

(e) Difference in Brain Alignment for Participant 8

Figure 14: Difference in brain alignment performance (measured by Pearson correlation) between brain brain-tuned and pretrained Wav2Vec2.0 models for different participants. It shows better alignment of the brain-tuned model in semantic language areas.

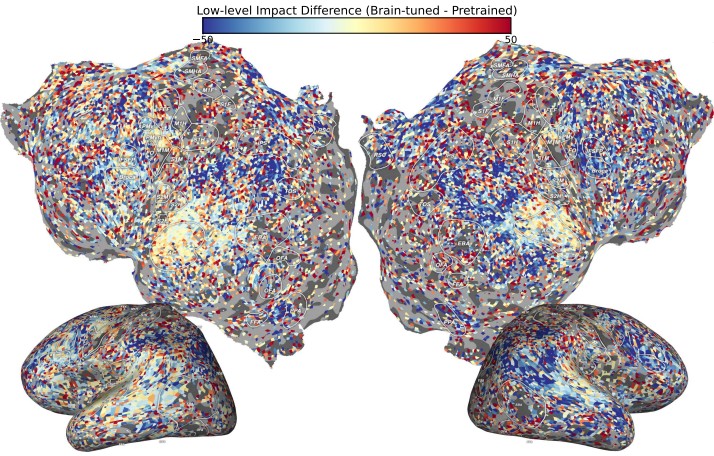

(a) Difference in low-level impact for Participant 1

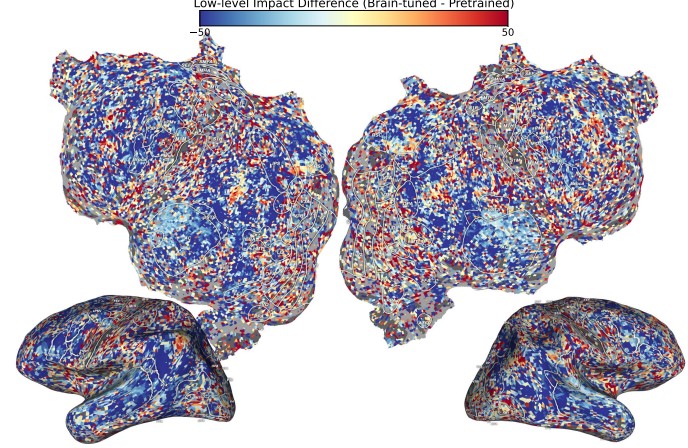

(b) Difference in low-level impact for Participant 2

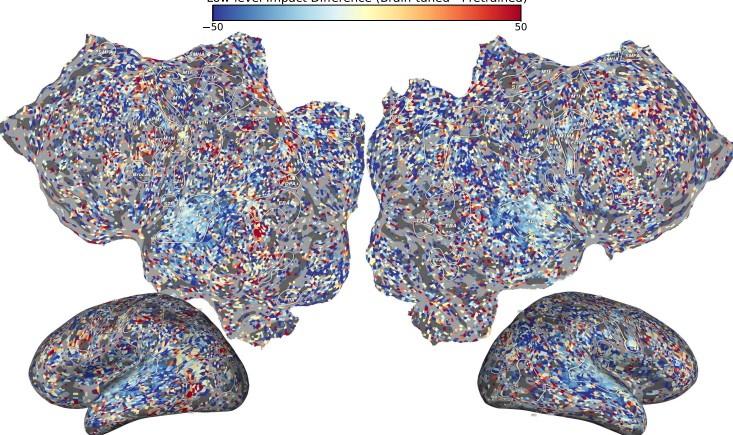

(c) Difference in low-level impact for Participant 6

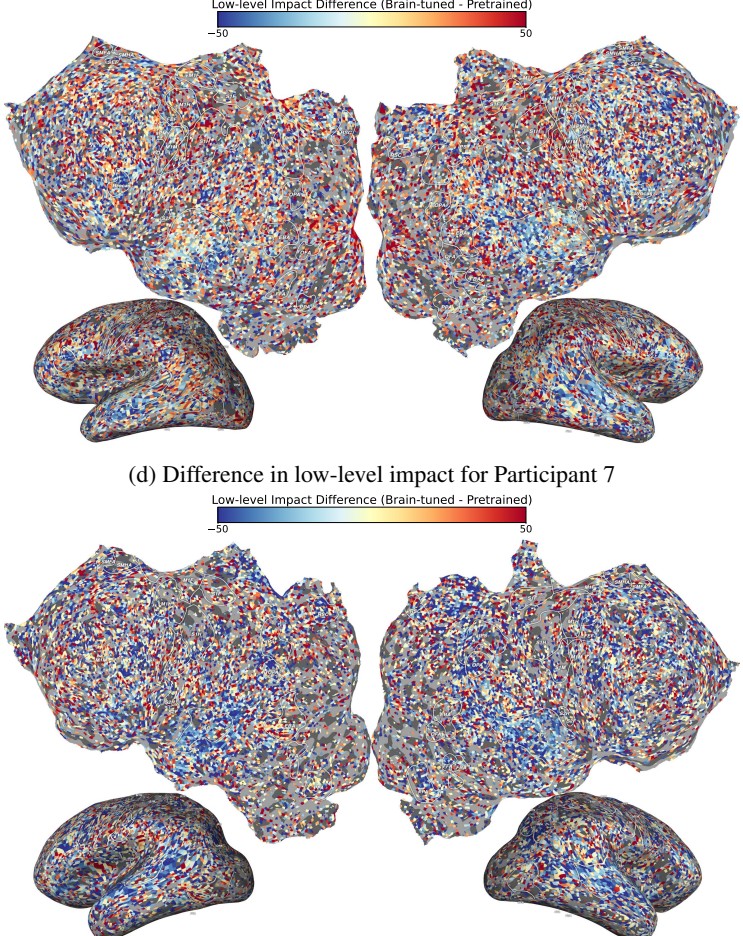

(d) Difference in low-level impact for Participant 7

(e) Difference in low-level impact for Participant 8

Figure 15: Difference in low-level impact brain brain-tuned and pretrained Wav2Vec2.0 models for different participants. It shows a lower low-level impact (lower drop due to low-level feature removal) for the brain-tuned model in semantic language areas.

