# OpenReview forum: "Improving Semantic Understanding in Speech Language Models via Brain-tuning"
_ICLR.cc/2025/Conference — ICLR 2025 Poster_

### Official Review · Reviewer_61Sy · 2024-10-17

**Soundness:** 3
**Presentation:** 4
**Contribution:** 3
**Rating:** 6
**Confidence:** 4

**Summary:**

The authors propose the concept of brain-tuning in this paper, which aims to augment pretrained speech model training directly with brain recordings. In this way, the authors claim to compensate for the shortcomings of previous methods that only use low-level speech features for brain and speech model alignment, and successfully incorporate speech models with improved brain-relevant semantics. Experiments on three popular pretrained transformer-based speech language model families in three distinct ways (alignment with new brain recordings in semantic regions of the brain, effect of low-level features, such as Tri-Phones and Articulation, on the alignment with these semantic regions, downstream performance on linguistic tasks that require semantic understanding) also highlight the significance of inducing brain-relevant semantics in brain and speech model alignment.

**Strengths:**

1. This paper first proposes fine-tuning pretrained speech models with brain responses to help improve the alignment between computational models and human brain. The experimental results are also impressive and solid.
2. The authors design several interesting experimental settings and baseline methods to help investigate the influence of brain-tuning.
3. The motivation is clear enough. The paper is also very well written with many experimental details supported.

**Weaknesses:**

1. Despite the authors effort in illustrating the effect of brain-tuning, the experiments are still insufficient in the following two aspects. (1)  The authors choose LeBel's dataset which only contains 8 subjects for practice. I don't think the conclusion drawn from this dataset is convincing enough given the very few subjects. Instead I suggest trying the Narratives dataset [1] which contains hundreds of subjects in a passive storying hearing task for experiments (by the way this dataset is considered as the largest fMRI dataset in naturalistic language comprehension, not the LeBel's dataset as the authors claimed in line 54), as many previous work [2][3][4] did. (2) The authors fail to investigate the influence of model parameters in the brain-tuning process. I think such experiments are easy but important, and should be included in the paper.

2. The methods of obtaining four low-level speech features need to be detailed in this paper (at least in Appendix), instead of just citing previous work.

[1] The “Narratives” fMRI dataset for evaluating models of naturalistic language comprehension. Nature Scientific Data

[2] Evidence of a predictive coding hierarchy in the human brain listening to speech. Nature Human Behavior

[3] Disentangling syntax and semantics in the brain with deep networks. ICML

[4] Deep language algorithms predict semantic comprehension from brain activity. Nature Scientific Reports

**Questions:**

1. Why is brain-tuning not effective for whisper? More analysis needs to be conducted. Is it caused by the encoder-decoder architecture?
2. How is the cortical parcellation conducted? The LeBel dataset doesn't provide any standardized space for all the subjects if I remember correctly. Are the fMRI images from different subjects first projected to the same space?

---

> ### Author Response · Authors · 2024-11-23
>
> Thank you Reviewer 61Sy for your detailed review of the work. We are happy you found the paper to be clearly written and the results to be solid and impressive. Below, we address the questions and concerns you have raised, which we hope increase the clarity and credibility of the work.
>
> 1. **fMRI Dataset Size**: The brain-tuning in our work trains one model for each participant. Therefore, the main criterion for choosing the dataset was the unique amount of data available for each participant, which is the largest for the LeBel's dataset (6.4 hours or more vs 4.6 for the Narratives dataset). Additionally, we repeat this per-participant brain-tuning over enough participants to show statistically significant results in brain alignment. The point of the work is that here we show that even data from one participant, which is equivalent to 0.7% of pretraining data of Wav2Vec2.0 is sufficient to improve semantic understanding in the models. We plan to extend the method to be applied across different participants in future work, which will be an ideal setting for the Narratives dataset that has many more participants.
> 2. **Influence of model parameters**: We are not certain we fully understand what the reviewer means by “influence of model parameters“, but we assume it to be the influence of model size (# parameters). We note that we leave brain-tuning larger models for future work, when the methods can incorporate brain data from multiple experiments in order to have more data. Still, brain-tuning smaller models is beneficial, and we further show that brain-tuning can bridge the gap in performance between smaller and larger models as shown in Appendix A.2 and E.2 (i.e., the performance of brain-tuned smaller models gets much closer to the performance of the pretrained larger models).
>
>
> 3. **Details on Low-level Speech Features**: We added more details about the definition and the extraction method of the low-level speech features in Appendix A.3.
> 4. **ًWhisper Performance**: Our results show that brain-tuning improves Whisper’s semantic understanding because 1) its downstream performance on more semantic tasks is improved and 2) the impact of low-level features on its brain alignment is decreased. As for the brain alignment results, please note that the results reported are not only for the voxels used during fine-tuning but for a wider set of voxels, which might explain the final brain alignment not necessarily increasing. That’s why, to more comprehensively assess the changes in a model thoroughly, we rely on multiple types of evaluation.
>
> 5. **Cortical Parcellation Method**: To extract language and speech regions, the cortical parcellation is indeed conducted by projecting the participant data to the same space. We use the FAverage standard surface version 7. [1, 2]
> Thank you very much for your help in evaluating our work! We hope we have addressed your questions and look forward to continuing the discussion if there are unresolved issues.
>
> [1] Bruce Fischl, Martin I. Sereno, Roger B.H. Tootell, and Anders M. Dale. High-resolution intersubject averaging and a coordinate system for the cortical surface. Human Brain Mapping, 8(4):272–284, 1999.
>
> [2] https://surfer.nmr.mgh.harvard.edu/fswiki/FsAverage

---

> > ### Comment · Reviewer_61Sy · 2024-11-24
> >
> > Thanks for the clarification of the authors. Since no additional results on narratives dataset and brain-tuning of large models are provided, I will retain my initial score.

---

> > > ### Author Response · Authors · 2024-11-28
> > > **New Results for Brain-tuning for increasing model sizes and amounts of fMRI data**
> > >
> > > We are happy to report that we were able to conduct new experiments that now show the effect of pretrained model size and amount of data on brain-tuning. We hope that you will take these new results into consideration and, with the extension of the Discussion period, we'll be able to hear your thoughts.
> > >
> > > **New results on brain-tuning pretrained models of increasing sizes:** We provide new results now in Fig 14a (Appendix E.3) for brain-tuned models in the HuBERT family of 3 sizes: 90M (the same one as in the main paper), 317M, and 1B. Due to time constraints, we were able to evaluate one model family and the effect on performance of 3 downstream tasks, but we see consistent results which increases our confidence in the results. In Fig 14a, we observe that brain-tuning all 3 model sizes results in increase in performance from the pretrained model of the corresponding size (e.g. brain-tuned HuBERT large > pretrained HuBERT large). However, as expected, this effect decreases with increasing model sizes (i.e. brain-tuning using the same amount of data is most effective for smaller models).
> > >
> > > **New results on the effect of the amount of fMRI data for brain-tuning:** We also now explicitly test the impact of the amount of brain data used during brain-tuning for the observed downstream effects in Fig 14b. Here we use the smallest size pretrained model. We observe a very clear increasing trend when scaling up the amount of brain data used during brain-tuning even of the smallest model. This provides clear evidence that data size plays a key role in the positive results we see from brain-tuning. Moreover, the increased trend has not leveled out yet, suggesting that incorporating more brain data during brain-tuning will yield further benefits. Unfortunately, no other public fMRI dataset contains more fMRI data per participant, as we described in the original response. In comparison to the current fMRI dataset we use, the largest Narratives dataset in terms of data per participant will come at around 71% of the data in our current setup. Therefore no current dataset is better suited to use in our brain-tuning method, and future methods that can combine data from multiple participants or multiple experiments may improve on our results.
> > >
> > > We'll add pointers to these results in the discussion and update the conclusion / future work with this evidence. We hope these new results address the reviewer's remaining questions.

---

### Official Review · Reviewer_SHAj · 2024-10-20

**Soundness:** 4
**Presentation:** 3
**Contribution:** 4
**Rating:** 8
**Confidence:** 4

**Summary:**

This paper explores an innovative approach to enhance the semantic alignment of speech language models with the human brain by incorporating brain data. By fine-tuning models using fMRI data of participants listening to natural stories, The goal is to improve the models' semantic understanding by aligning them better with brain regions involved in language comprehension. The authors evaluate the method across three model families (Wav2vec2.0, HuBERT, Whisper) and show improvements in brain alignment, reduced reliance on low-level features, and better performance on downstream tasks that require semantic understanding. This idea is excitingly novel due to its cross-disciplinary implications between neuroscience and AI.

**Strengths:**

1. **Novelty**: The concept of "brain-tuning" is a novel way to integrate neuroscience data into AI training, bridging the gap between machine learning and brain sciences.
2. **Comprehensive Evaluation**: The evaluation strategy is thorough, with alignment tests, low-level feature analysis, and performance on multiple downstream tasks.
3. **Cross-disciplinary impact**: The paper offers insights valuable to both cognitive neuroscience and AI communities, particularly for understanding semantic processing in the brain and improving language models.
4. **Significant Results**: The results are robust, showing improved alignment with brain regions and downstream task performance, indicating the model's improved semantic capabilities, **but only in their restricted comparison models.**

**Weaknesses:**

1. **Comparison models are too weak and restrictive for the experiment**: There are **only two** comparison models: BigSLM Fine-tuned and Random-fMRI, the latter is not very valid since using random-fMRI results will definitely deteriorate performance, so there is in fact **only one** valid compasion model BigSLM, which is not convincing at all! BTW, BigSLM sometimes degrade performance as Fig 4, it's not valid as well.  **To prove effectiveness of brain-tuning, the paper should better compare brain-tuning with other valid finetuning models, ont only pretrained models.**
2. fMRI data, in some degree, is too hard to acquire. As the paper mentioned, the largest public dataset of fMRI recordings only cover 8 participants. This will limit the usability of the proposed method, but this won't downgrade my rating for this paper since I think the idea side is novel and useful enough.

**Questions:**

1. Please provide more detailed explanation why so few comparison models are provided or add more comparison models in the paper. This will dramatically affect my rating, if no reasonable explanation were provided, I would highly question the effectiveness of the method and definitely downgrade my rating.
2. If possible, answer the second weaknesses, this is relatively subtle.

---

> ### Author Response · Authors · 2024-11-23
> **New LM-finetuned baseline and clarifications**
>
> Thank you Reviewer SHAj for your attentive review of the work. We are happy you find the work novel, thorough, and significant. Below, we address the concerns you have raised, which we hope increase the clarity and quality of the work:
>
> 1. **New baseline and clarification of existing baselines**:
>
>     1. **LM fine-tuned baseline**: we now implement a new baseline to alleviate your concerns. We fine-tune each pretrained model using representations from a text-based language model (GPT2), and show that this baseline also performs worse than the brain-tuned models (Appendix D.2). Please see point 4 in the common response for more details.
>
>     2. **BigSLM fine-tuned baseline**: we believe a BigSLM Baseline is the best baseline to understand the added benefit of fMRI data over speech models, which is the aim of our approach.  We also note that this baseline outperforms the pretrained model or performs on par with it for most tasks (Fig 4 in the updated paper).
>
>     3. **Random-fMRI baseline**: we believe this baseline is important as it’s not always true that random or noisy targets are trivially insignificant to the model’s performance. As we detail in the common reply (Point 4.3), sometimes they act as regularization and can help the model. Hence, it’s important to add a grounding baseline to compare the fMRI against. ‌  ‌
>
>       Based on the points above, we think that we covered important comparison models that provide both semantic and speech improvement comparisons against the brain-tuning models. We have also shown that their behavior is consistent across 3 model architectures. We hope this clarifies the concerns about the effectiveness of the comparison models we used, and please let us know if you have more thoughts on other baselines.
>
>
> 2. **Brain Data Limitation**: The concern about data size and collection is of course valid. The benefit of the current work is that here we show that even data from one participant, which is equivalent to 0.7% of pretraining data size of Wav2Vec2.0 is sufficient to improve semantic understanding. In the future, our method can be extended to incorporate data from multiple experiments ( multiple participants and datasets). This may also enable the application of brain-tuning to larger models.
>
> Thank you very much for your help in evaluating our work! We hope we have addressed your questions and look forward to continuing the discussion if there are unresolved issues.

---

> > ### Comment · Reviewer_SHAj · 2024-11-25
> >
> > 1. Is the evaluation dataset more similar to the fMRI audio data than to the pretrained audio data? This distinction could significantly influence the results and should be carefully considered.
> >
> > 2. The model used in BigSLM (Whisper Medium) is relatively small and somewhat outdated, as is GPT-2. Given the computational demands of this scenario are primarily tied to inference, employing a larger and more recent model would not introduce a substantial computational burden and may yield better results.
> >
> > 3. As you said, random fMRI acts as random or noisy targets, only *sometimes* act as regularization. Comparison with random fMRI only shows superiority over random or noisy targets, it does not serve as a valid fine-tuning method.
> >
> > The work is around the acceptance threshold, I may retain my original score very likely.

---

> > > ### Author Response · Authors · 2024-11-26
> > >
> > > Thanks for engaging in a discussion.
> > >
> > > 1. The datasets for evaluation on downstream tasks, pretraining, and fMRI audio are all different. We don't believe the evaluation data is necessarily close to either the pretraining or to the fMRI audio. The evaluation datasets are composed of independent sentences that are quite controlled and can deviate from natural speech. The pretraining data (e.g. LibriSpeech) contains hundreds of speakers reading excerpts from audiobooks [1]. The fMRI audio is short stories from a real radio podcast (Moth Radio Hour on NPR). The fMRI audio is the most natural of all datasets.
> > > 2. We want to clarify that the computational demands in our method are coming exactly from the fine-tuning step and not from the inference. Using larger models as targets during fine-tuning will substantially increase the computational demands (e.g. Llama-70B has 80 layers with each having embeddings of size 8192, which means that the fine-tuning targets will be ˜650k, which is more than a 20x increase from our current setup). The inference demands are the same, regardless of what model was used for fine-tuning, because the base model is the same. We also want to clarify that Whisper Medium has 769M  parameters, and Whisper is on par with state of the art and is still widely used in multilingual ASR[2, 3]. The reason why we chose GPT2 for the LM-finetuning is that it has a similar size to Whisper Medium.
> > > 3. We don't suggest random-fMRI as a method that should be used by others -- we only include it to show that the same benefit from brain-tuning cannot be gained via simple regularization due to noisy targets. We think this increases the confidence in the brain-tuned results, and Reviewer x8dp specifically points this out as well.
> > >
> > > We also included a new baseline in the first response, as suggested by Reviewer BDuH: an LM-finetuned baseline. If you have suggestions for other types of baselines that would be useful to run and would not require magnitudes more compute, we would be happy to implement this in the remainder of the discussion period.
> > >
> > >
> > > [1]  "LibriSpeech: an ASR corpus based on public domain audio books", Vassil Panayotov, Guoguo Chen, Daniel Povey and Sanjeev Khudanpur, ICASSP 2015.
> > >
> > > [2]Song, Z., Zhuo, J., Yang, Y., Ma, Z., Zhang, S., & Chen, X. (2024). LoRA-Whisper: Parameter-Efficient and Extensible Multilingual ASR. arXiv preprint arXiv:2406.06619.
> > >
> > > [3] Peng, J., Wang, Y., Xi, Y., Li, X., & Yu, K. (2024). A Survey on Speech Large Language Models. arXiv preprint arXiv:2410.18908.

---

> > > > ### Comment · Reviewer_SHAj · 2024-12-02
> > > >
> > > > 1. A better idea would be adding another model which pretraining dataset including fMRI audio. This would be a lot convincing for you method.
> > > > 2. GPT-2 is not only small but also old, if Llama-70b is too big, you can use Llama-7b, StableLM, Phi-2 etc. this would a lot better. Thanks for pointing out my mistake about compution, but I think you may use embedding of, like, every 3 layers.
> > > > 3. You response is generally consent to mine, this doesn't count a good comparsion model.
> > > >
> > > > I‘m thinking about raising my score to 8. But I really wanna you think about point 1 and 2.

---

> > > > > ### Author Response · Authors · 2024-12-02
> > > > >
> > > > > Thank you for considering our response carefully and for continuing to engage in a discussion. Thanks also for the suggestions for new comparison models. We believe these can further strengthen the confidence in our results. We'll work on incorporating these suggestions in the final version of the manuscript if it is accepted. Below we detail the specific experiments we will deliver and our expectations for their results:
> > > > >
> > > > > 1. If we understand correctly, the suggestion is to compare our brain-tuned model (which is pretrained on data X and then fine-tuned with pairs of fMRI stimulus audio and fMRI data) with a model that is pretrained on X + the fMRI stimulus audio. Then any difference in performance will be due to only the added fMRI data without the addition of new fMRI stimulus audio, because both models will have seen exactly the same inputs. We agree this is a good baseline. Our BigSLM fine-tuned baseline accounts for the additional data related to the fMRI stimulus audio, as the BigSLM fine-tuned and brain-tuned models are trained on the exact same inputs, which makes us confident that the brain-tuned model does not only outperform the pretrained because it has seen more audio data. However, we agree that your suggestion will more clearly show the benefit of fMRI data. We expect this new baseline to perform worse than the BigSLM fine-tuned baseline, since both the new baseline and the BigSLM baseline see the same input data but the BigSLM fine-tuned baseline adds additional information from the big speech model. Therefore, we expect that the new baseline will also perform worse than the brain-tuned model on all semantic tasks, where brain-tuned already outperforms BigSLM fine-tuned (Fig 4). We will implement this new baseline by training the pretrained models with their original pretraining objective only on the fMRI stimulus audio and will add the downstream performance of this model to Fig. 10.
> > > > >
> > > > >
> > > > > 2. We'll provide downstream tasks results from an LM-finetuned baseline using representations from Llama2 7b as a fine-tuning target and add them to Fig. 10. We expect that this model will outperform the GPT2 fine-tuned baseline on the tasks where we observe that LM fine-tuning is helpful (e.g. GPT2-finetuned > BigSLM-finetuned in Fig 10). This will provide additional evidence that improved semantics also improves performance on these tasks. It is an interesting question how the Llama2 fine-tuned model will compare to the brain-tuned on these tasks. Even if the Llama2 fine-tuned model outperforms the brain-tuned model on these tasks, the brain-tuned performance on these semantic tasks, compared to the BigSLM fine-tuned and pretrained, is still noteworthy, as it shows that fMRI signals can be used to improve semantics in speech models, which is the main aim of our work. On the remaining tasks that are less semantic, we do not expect that the new Llama2-finetuned baseline will perform better than the GPT2-finetuned baseline. Therefore, we expect the Llama2-finetuned baseline to perform substantially worse than the BigSLM-finetuned and brain-tuned models on these less semantic tasks.

---

> > > > > > ### Comment · Reviewer_SHAj · 2024-12-03
> > > > > >
> > > > > > 1.  Adding a group pretrained on X + the stimulus fMRI audio is crucial. It's much more reasonable:
> > > > > >     - Although BigSLM model have seen exactly the same inputs as brain-tuned model, it acts substantially worse than the base model on **not a few** metrics and dataset. And it doen't count as a basic control group from the perspctive of method.
> > > > > >     - It's unfair to compare a model only pretrained on fMRI stimulus audio with brain-tuned model which is pretrained on X and brain-tuned on fMRI stimulus audio. This comparison cannot show superiority of brain-tuned method **at all**.
> > > > > > 2. I generally agree on your last expection.

---

> > > > > > > ### Author Response · Authors · 2024-12-03
> > > > > > >
> > > > > > > Great, it sounds like we understood your suggestions correctly. Just one clarification on the BigSLM results: the BigSLM fine-tuned model performs on par or better than the pretrained model in 16 of the 17 comparisons (Fig 4). The only setting in which the BigSLM fine-tuned model performs substantially worse than the pretrained model is on sequence understanding for HuBERT (Fig 4c, middle bar), where the performances are already very low across the board.
> > > > > > >
> > > > > > > We’ll deliver both baselines 1 and 2 that you suggested for the final version of the manuscript if it is accepted. We really appreciate your help in strengthening our work and will be thankful if you reflect your support by increasing your score.

---

> ### Comment · Reviewer_SHAj · 2024-12-03
>
> Thanks, I believe this two new baselines will make the work substantially convincing and I've updatied the score.

---

> > ### Author Response · Authors · 2024-12-03
> >
> > We greatly appreciate the reviewer’s responsiveness, detailed insights, and support reflected in the upgraded score.

---

### Official Review · Reviewer_BDuH · 2024-11-03

**Soundness:** 3
**Presentation:** 3
**Contribution:** 3
**Rating:** 6
**Confidence:** 4

**Summary:**

The paper incorporates a new finetuning approach for speech encoder models using neural recordings from subjects listening to stories. The approach uses the fMRI responses as a target during finetuning and modifies transformer layers of the speech. The paper demonstrates three areas of general improvement: (1) better brain alignment, (2) reduced impact of low level features for brain alignment, and (3) improved downstream performance across certain tasks.

**Strengths:**

* Novelty: The paper isn’t the first to propose brain-tuning (which has been proposed in the vision and language domains) but is the first to apply this approach to speech models which often lack semantic understanding.
* Improvements in downstream performance: Most impressively is the finding that tuning with fMRI leads to strong improvements in ASR. While prior work has shown improvements from incorporating neural signals, I haven’t seen consistent improvements on downstream performance.
* Reduced impact of low level features in brain alignment: Low level features become less impactful indicating fewer simple audio features are driving alignment to the brain. This is a beneficial and impactful finding.
* Phonetic-Semantic Split: Following the prior strength, I liked Appendix C which quantified the phonetic-semantic preferences between finetuning and pretrained models.

**Weaknesses:**

* Baselines: In general, I think the current baselines are noteworthy but I think there are a few baselines which are missing.
    * If the goal of the paper is to improve semantic understanding in speech models, the SLM baseline is a bit odd to me. Wouldn’t it be better to use a pretrained language model embedding as a target instead? I would recommend adding this baseline as another way to incorporate semantic information into the model. I also found a model embedding target to be a bit unfair in comparison to fitting voxels since the target is one-dimensional but any target from an embedding model is higher-dimensional according to my understanding, reducing the complexity of the fit.
    * I also think it would be great to include a baseline that controls for particular voxels of fMRI recordings. For example, I was wondering whether the area of the brain had an effect on the outcome of the model.
* Forgetting: One problematic aspect of finetuning is catastrophic forgetting where models can forget currently learned capabilities. I have this concern after seeing the performance decrease in your emotion prediction task. It would be very useful to quantify this point more formally to give a sense of whether semantic information is better incorporated at the sacrifice of basic phonetic understanding. This doesn’t seem to be reflected in appendix C as well. I would appreciate seeing 1-2 low level audio tasks benchmarked as well.
* Neural Signal Details: Some details were missing. See questions.
* (Nit) Consider adding [1] to your related work as another paper that tunes models based on neural information.

Overall, I enjoyed this paper. I think adding a few additional baselines will make this paper much stronger. After seeing these, I will raise my score.

[1] Dapello et. al. 2023 Aligning Model and Macaque Inferior Temporal Cortex Representations Improves Model-to-Human Behavioral Alignment and Adversarial Robustness. ICLR 2023

**Questions:**

* What areas of the brain were voxels covering for your fMRI target? Were semantic areas covered or other areas covered?
* My familiarity with fMRI is low by why would you downsampling using a 3-lobed Lanczos filter? Just a bit confused on the need to do so, not anything wrong with the choice.

---

> ### Author Response · Authors · 2024-11-23
> **New LM-finetuned baseline, new results on brain-tuning with specific brain regions, new results on low-level speech tasks, and clarifications**
>
> Thank you Reviewer BDuH for your in-depth review of the work. We are pleased you enjoyed the paper and found our approach and results impactful and beneficial. Also, thank you for referring us to the work of Dapello et. al. 2023 in the vision domain, which we will refer to in the next revision of the manuscript. Below, we address the raised concerns:
>
> 1. **LM-finetuned baseline**: Please see point 4.1 of the common response above. Briefly, we have added an LM (GPT2)-finetuned baseline, and we report its results on downstream tasks against brain-tuned models in Appendix D.2. Our brain-tuned model consistently outperforms this LM baseline.
> 2. **BigSLM Baseline**: Please see the common response point 4.2 for more details. Briefly, we believe a BigSLM Baseline is the best baseline to understand the added benefit of fMRI data over speech models, which is the aim of our approach.
> 3. **Dimensionality of Targets**:  We clarify that the BigSLM targets and the fMRI data are of similar dimensions: ~30-40K dimensional vector for the fMRI targets and ~26K for the BigSLM targets). Both fMRI targets and BigSLM targets are also treated similarly in the MSE loss as it is applied to the flattened high-dimensional vector of both in the same way, not per voxel for the fMRI case (all used voxels are flattened into a vector and used to obtain one MSE value). Hence, the way we apply the loss and the complexity of the reconstruction is comparable in both cases.
> 4. **Brain areas covered in brain-tuning**: Based on qualitative analyses, our brain-tuning approach covers a large number of semantic regions, as well as the auditory cortex (Appendix F.1). Your suggestion to brain-tune with specific regions is very interesting; however, only a couple of regions (namely Broca and Auditory Cortex (AC)) are anatomically annotated across subjects by the authors of the datasets, limiting our ability to do a thorough ablation study on the different regions/ networks of the brain. Still, we did initial experiments with brain-tuning Wav2vec2.0 only using these two regions (Broca and AC). We include the results in Appendix E.1 of the paper, which show that brain-tuning using these two regions boosts the downstream performance compared to the pretrained model, but it still underperforms compared to using the noise-filtered whole brain. Trying to efficiently post-process the original dataset to be able to functionally localize all relevant semantic and speech regions is one of the future directions.
> 5. **Concern about potential forgetting of speech capabilities**: Please take a look at point 3 in the common response. Briefly, we believe our brain-tuning setup avoids catastrophic forgetting of the speech capabilities because the brain regions that are included in the training do not only include semantic regions, but also low-level speech regions (the auditory cortex). Furthermore, we now show results for 2 additional low-level tasks (MFCC and FBANK prediction), further supporting that brain-tuned models maintain speech capabilities (Appendix D.1).
> 6. **3-lobed Lanczos filter**: This filter acts as a low-pass antialiasing filter (with a cut-off at the Nyquist frequency of the fMRI) and allows for better and more efficient downsampling of the convolved input signal (the input signal after applying the filter). Moreover, it’s been shown to be consistently better than applying an average filter to downsample the input [2], which is why it's a standard choice in the literature [1,2].
>
>
> Again, thank you very much for your detailed and clear review! We hope we have addressed your questions and look forward to continuing the discussion if there are unresolved issues.
>
>
> [1​​] Oota, S.R., cCelik, E., Deniz, F., & Toneva, M. (2023). Speech language models lack important brain-relevant semantics. Annual Meeting of the Association for Computational Linguistics.
>
> [2] LeBel, A., Wagner, L., Jain, S. et al. A natural language fMRI dataset for voxelwise encoding models. Sci Data 10, 555 (2023). https://doi.org/10.1038/s41597-023-02437-z

---

> ### Comment · Reviewer_BDuH · 2024-11-25
>
> I thank the authors for additional clarifications and baselines. However, I am left a bit confused about the reported result from using an LM representation as a target as requested. The authors claim that they can remove low level features and improve semantic understanding in speech encoder models. Yet they do so with fMRI, an incredibly noisy target. When they replace this target with a GPT-2 embedding -- a more feature rich, clean representation of semantics -- the result is worse? In fact, if I am reading the figure correctly, it performs even worse than the base model at times. I had initially explained the result from the BigSLM as follows: the audio model as a target was reinforcing particular features, causing less robust improvement. This would have been complemented by using an LLM target. But now the explanation seems inadequate.
>
> I would really appreciate if the authors could provide some discussion on this result and their intuitions for what is happening. My sense is that the language model should be an upper bound for a clean representation of semantics. That intuition could be wrong, but I find myself a bit surprised by the finding.

---

> > ### Author Response · Authors · 2024-11-25
> >
> > We agree that some of these new results are surprising. Thanks for the opportunity to discuss further. Our intuition behind why the LM-finetuned model is not serving as an upper bound on performance for these tasks is two-fold:
> >
> > **LM-finetuned vs BigSLM-finetuned:** As we discuss in the paper, previous work has shed doubts on the degree of semantics that some downstream tasks require. We observe that both the BigSLM-finetuned and pretrained models outperform LM-finetuned in 2 of the 4 tasks in Fig. 10 (Appendix D.2), which suggests that these two tasks may also need some phonetic and low-level speech features, a finding that’s also supported by [1].  Therefore, we think that the decrease of lower-level speech capabilities in the LM-finetuned model (which is evident from the new evaluation results on low-level speech tasks we present in Fig. 11) plays a role in the results we observe in Fig.10.
> >
> > **LM-finetuned vs brain-tuned:** We believe our brain-tuned models outperform the LM-finetuned on all of these 4 tasks for two reasons:
> >
> > 1. Firstly, brain-tuning does not decrease lower-level speech capabilities as much as LM-finetuning does (Fig. 11). As we discuss in our first common response, we believe this is because the fMRI signals we include in the brain-tuning are from both semantic and lower-level auditory regions. Combined with the reasoning above about the necessity of semantics for the tasks, we believe this is the main reason that the brain-tuned model outperforms the LM-finetuned by a large margin on 2 of the 4 tasks (essentially, because the LM-finetuned is dropping in performance while the brain-tuned model maintains performance).
> >
> > 2. Second, brain-tuning actually improves semantics beyond GPT2. We observe that the two tasks where LM-finetuned > BigSLM are also where the brain-tuning models have the biggest gains over the rest of the baselines, including LM-finetuned (Fig 10). We agree that this result is surprising. Our intuition behind this result is the following: humans outperform GPT2 on many semantic tasks, so human brains must clearly have better semantic understanding than GPT2. Now the question is whether this better semantic understanding is reflected in fMRI signals. We are not aware of any work that has shown whether these task-relevant semantics are present or absent from fMRI signals. Our work provides empirical evidence that these semantics are indeed present in the fMRI signal and are useful for improving performance on semantic tasks.
> >
> > We will include this discussion in the paper.
> >
> > We also want to highlight here that the point of our work is not to create the most semantically advanced speech model. Our aim is to improve brain-relevant semantics in the speech models so they can serve as better model organisms of listening in the brain. The fact that these improved brain-relevant semantics also improve performance on downstream semantic tasks is a byproduct. There may be other ways to improve downstream task performance in models, as suggested by the reviewer. Still, the fact that brain-tuning is able to supply this signal is noteworthy.
> >
> > We hope that the reviewer can take this result and discussion, as well as the additional results we provided in the previous response following their suggestions (new results on brain-tuning with specific brain regions and new results on low-level speech tasks) into account when evaluating our work.
> >
> > [1] Choi, K., Pasad, A., Nakamura, T., Fukayama, S., Livescu, K., & Watanabe, S. (2024). Self-Supervised Speech Representations are More Phonetic than Semantic. arXiv preprint arXiv:2406.08619.

---

> > > ### Comment · Reviewer_BDuH · 2024-11-26
> > >
> > > Thanks for providing some intuition on the results. I really appreciate the authors taking the time to engage with my concerns. I want to comment that I don't feel that the other contributions of this paper don't count. My aim here is to understand the reported improvement on downstream performance with the context of this new baseline.
> > >
> > > I feel like this result makes me confused about what the fMRI is representing that is useful for the speech model. The authors present an intuition that it balances both semantic and phonetic information. However, that may not be useful for a task like ASR. Would the results in the paper imply that you could linearly/non-linearly decode audio/speech features directly from fMRI? For example, if I were to believe that fMRI was useful for an audio model, then I should be able to decode ASR/phonemes/etc. with better performance than decoding from the pretrained model. Or at least that information should be encoded in fMRI somewhere and should be decodable at an accuracy that is above chance. Could the authors provide some discussion on this intuition?

---

> > > > ### Author Response · Authors · 2024-11-26
> > > >
> > > > We really appreciate your engagement! Low-level speech features have indeed been shown to be encoded in fMRI signals, specifically in the auditory regions [1,2,3]. This is why we believe that our method which includes signals from the auditory regions, as well as semantic regions, during brain-tuning is able to maintain the speech models’ speech capabilities.
> > > >
> > > > We don’t expect that fMRI encodes more accurate low-level speech information than speech models already have. This is also supported by the new results we provided on low-level tasks (Fig 11), where the brain-tuned model performs on par with the BigSLM model. Similarly, we see that for tasks like ASR, the difference between the BigSLM baseline and the brain-tuned model are quite small (Fig 4a). Importantly though, we show that the brain-tuning does not substantially reduce these low-level speech capabilities either (unlike LM-finetuning).
> > > >
> > > > What we believe is most useful for the speech model from fMRI signals is semantics. It’s difficult to prove this but there are two lines of evidence that support this conclusion: 1) we show that the impact of low-level features on the alignment with **semantic** brain regions is significantly reduced (Fig 3, also see response to Reviewer x8dp on “Low-level Impact Reasoning”), and 2) the tasks where we see the biggest gains from brain-tuning over the BigSLM and pretrained baselines are the same ones for which LM-finetuning also outperforms the other baselines (Fig 11), indicating that these may be the tasks where better semantics improves performance. Note that we have not been able to evaluate the LM-finetuned model on ASR due to time constraints but we don’t expect it to improve on the other baselines, since we also see that the brain-tuning does not improve much on the BIgSLM model for ASR (Fig 4a). There is also evidence from neuroscience that semantic content can be decoded directly from fMRI responses of people listening to stories [4].
> > > >
> > > > We appreciate your push on this! We will add a discussion of our expectations for what the fMRI signals can improve and maintain in the speech models to the introduction and then discuss again in light of our results in the discussion. Thank you, we think this will help communicate the contributions of our work more clearly.
> > > >
> > > >
> > > >
> > > > [1] de Heer, W. A., Huth, A. G., Griffiths, T. L., Gallant, J. L., & Theunissen, F. E. The hierarchical cortical organization of human speech processing. Journal of Neuroscience, 2017.
> > > >
> > > > [2] Vaidya, A. R., Jain, S., & Huth, A. Self-Supervised Models of Audio Effectively Explain Human Cortical Responses to Speech. ICML, 2022.
> > > >
> > > > [3] Oota, SR, Çelik, E., Deniz, F., and Toneva, M. Speech language models lack important brain-relevant semantics. ACL, 2024.
> > > >
> > > > [4] Tang, J., LeBel, A., Jain, S., & Huth, A. G. Semantic reconstruction of continuous language from non-invasive brain recordings. Nature Neuroscience, 2023.

---

> > > > > ### Author Response · Authors · 2024-11-29
> > > > >
> > > > > We would like to add that we have updated the downstream results for the LM-finetuned baseline to have all tasks (App D.2 - Fig.10). Before we generated these results, we predicted that LM-finetuning won't be helpful on these tasks, since they're also not helped substantially by brain-tuning (i.e. the brain-tuning and BigSLM fine-tuned baseline are close). The added results match these expectations.
> > > > >
> > > > > We hope this helps increase the reviewer’s confidence in our results, and we are happy to answer any other questions.

---

> ### Comment · Reviewer_SHAj · 2024-11-26
>
> It's possible though. LM representation may not contain much low-level speech feature.
>
> Also note in Figure 10, authors only post partial comparison results, not as comprehensive as Figure 4.
>
> I also doubt that is the evaluation dataset more similar to the fMRI audio data than to the pretrained audio data.

---

> > ### Author Response · Authors · 2024-11-26
> >
> > We're not sure what the first line refers to and would appreciate a clarification.
> >
> > Regarding Fig 10: these results are on 4 of the 6 tasks shown in Fig 4. The two tasks that we have not included in Fig 10 are those that brain-tuning is not as helpful on even when compared to the existing baselines (BigSLM fine-tuned), so we omitted them from the LM fine-tuning evaluation due to time constraints. We don't expect the LM fine-tuning to help on those tasks, since we observe from the other 4 tasks, that the only tasks that LM fine-tuning is helpful on are those where brain-tuning outperforms the other baselines by a large margin (Fig 10).
> >
> > Regarding the last point: we answer this doubt now in your original response thread. Briefly, the fMRI audio is not closer to the evaluation datasets we used.

---

> > > ### Author Response · Authors · 2024-11-29
> > >
> > > We would like to bring to the reviewer’s attention that we managed to include the remaining two tasks for the LM-finetuned baseline in the reviewed manuscript. Before we generated these results, we predicted that LM-finetuning won't be helpful on these tasks, since they're also not helped substantially by brain-tuning (i.e. the brain-tuning and BigSLM fine-tuned baseline are close). The added results match these expectations. We believe this fully adds another solid comparison model that further strengthens the confidence in our results.

---

> > > > ### Comment · Reviewer_BDuH · 2024-12-01
> > > >
> > > > Thank you for the detailed response! I have carefully read over the responses and experiment. I think I have a better sense of the improvements from fMRI tuning and the authors have resolved my concerns about the LM finetuning result. And the other results are also very interesting. I appreciate the authors testing on the original dataset as the pretrained speech models as well. I will upgrade my score.
> > > >
> > > > It would be great if these discussions are able to make it into the final paper. And over time, I think it would be great to get a deeper answer on what changes in the speech model i.e. through some kind of probing analysis.

---

> > > > > ### Author Response · Authors · 2024-12-03
> > > > >
> > > > > We thank the reviewer for the careful reading of the added results and discussions, and for showing support by upgrading the score. We have included some discussions related to the LM fine-tuned baseline in the latest revision of the manuscript, and we will complete the edits we promised related to expectations about what fMRI signals can improve and maintain in speech models in the next version.

---

### Official Review · Reviewer_x8dp · 2024-11-04

**Soundness:** 3
**Presentation:** 3
**Contribution:** 2
**Rating:** 6
**Confidence:** 3

**Summary:**

- In short the main idea is to fine-tune existing speech models so they are better aligned with fMRI recordings and then show that this results in better, more semantic alignment with the brain, and better downstream performance on speech tasks.
- The data comes from fMRI recordings that have been aligned to the podcast audio stimulus. For baselines, the authors run the off-the-shelf models and also fine-tune using fMRI signal unrelated to the task as well as audio representations from another model
- Finally, the authors formalize a notion of low-level feature impact, i.e., how much of the alignment can be attributed to low level (non-semantic) audio features, and show that their brain-tuning results in an alignment that can be explained less by low-level features.

**Strengths:**

- The fact that brain-tuning yields improvements on automatic speech recognition is very noteworthy. I have some points that need clarification (see below) before I can endorse this conclusion, but if they are addressed, then this will increase my confidence in the significance of this work.
- The motivation for this project is good. This work closes the loop on the large amount of existing work that has shown the emergent alignment between pretrained speech models and the brain. It appears novel in this respect.
- The random fMRI and BigSLM baselines increase my confidence in the fact that the recorded brain activity is a useful target for alignment.

**Weaknesses:**

- The conclusion that brain-tuning yields improvements over the off-the-shelf models is incredible, to the degree that I feel I must be misinterpreting the conclusion. There are two things I want to be sure of before this conclusion can be accepted:
  - First, is there any architectural enhancement that the brain-tuned version of wav2vec2 has that the original version does not? Extra projection layers?
  - Second, are the performances of wav2vec2 and HuBERT being fairly represented? It seems puzzling to me that the reported off-the-shelf ASR performances are so low (see questions below). One thing that may be happening: there is a distribution difference between the Librispeech data that wav2vec2 is trained on and the podcast audio used in the experiment. During brain-tuning, the model gets a chance to fine-tune on the podcast audio, or a proxy for that audio (via the brain), and this explains the improved performance. The gold standard experiment for this claim would be to do brain-tuning over one dataset of fMRI-audio pairs, and then show improvement on a completely new set of audio. To this end, it would be good to report the model performance on their original tasks.
- The fact that brain-tuning does not result in catastrophic forgetting of the original speech model representations is so incredible that I feel I must be missing something. The brain is a very noisy proxy for the audio signal, and using the brain as a target seems to only give very weak incentive to preserve the original speech representations (see questions). As mentioned above, it would be good to report model performance on their original eval datasets (ex. Librispeech) to make sure that any forgetting did not occurr.

**Questions:**

- How is performance on the original speech task preserved even during brain-tuning? Do the authors have an explanation for why no catastrophic forgetting occurs, given that there is no incentive for the base model to retain it's original speech representations during brain-tuning? Is it just because the learning rate is very small? The brain-tuning set is very small?
- The ASR performance of wav2vec2 seems low? The original wav2vec2 paper (https://arxiv.org/pdf/2006.11477) reports accuracies of >0.8 on Librispeech. What explains the difference?
- I'm having a hard time following the reasoning behing the low-level feature impact results. If I understand correctly, models were brain-tuned. And then the low-level impact on the brain-tuned models was found to be smaller, compared to the random fMRI baseline. Why should this be the case? Shouldn't we expect the opposite based on the evidence cited on lines 237-238, which suggests that brain-tuning should increase alignment between the brain and the model, and therefore the reliance on low level features?

## small stuff
- Figure 1c: Are the bar charts supposed to have an x-axis label?
- Line 257: `` gives open quotes in LaTeX

---

> ### Author Response · Authors · 2024-11-23
> **New results on low-level speech tasks, and clarifications**
>
> Thank you Reviewer x8dp for your detailed reading and constructive criticism. We were happy you found the paper well-motivated and novel. Below, we address the questions and concerns you have raised, which we believe improve the clarity and strengthen the confidence in the results:
>
> 1. **Architecture:** The only addition to the speech model for brain-tuning is a linear projection layer from the output of the speech model to the fMRI signal (similar to any fine-tuning pipeline). There are no additional architectural changes to the original version of each model.
>
> 2. **Evaluation audio datasets**: All datasets used for downstream evaluation were not seen before by any of the models. The datasets are completely new audio, different from the fMRI audio data used for brain-tuning as well. We believe this ensures fairness in the evaluation. Please refer to the common reply point 2 for further details.
>
> 3. **ASR performance**: Thank you for the careful reading of the methods and results. We want to clarify two points concerning ASR results.
>
>     - The ASR performance we report in the revised Fig 4a is on par with previous results on the TIMIT dataset. There was a plotting mistake in the original subplot, which made the ASR performance appear low. The correct results were reported in the text, and we have adjusted the plot to correctly reflect these results. Please refer to point 1 of the common reply above for more details.
>
>     - The reason why our reported ASR results are on TIMIT and not on the Librispeech dataset, as in the original Wav2Vec2.0 paper, is that we wanted to use a dataset for evaluation that did not unfairly bias the evaluation towards either the pretrained or the brain-tuned model (Wav2Vec2.0 is pretrained on Librispeech).
>
> 4. **Concern about potential catastrophic forgetting of speech capabilities**: Please take a look at point 3 in the common response. Briefly, we believe our brain-tuning setup avoids catastrophic forgetting of the speech capabilities because the brain regions that are included in the training do not only include semantic regions, but also low-level speech regions (the auditory cortex). Furthermore, we now show results for 2 additional low-level tasks (MFCC and FBANK prediction), further supporting that brain-tuned models maintain speech capabilities (Appendix D.1).
>
> 5. **Low-level Impact Reasoning**: The work discussed in lines 237-238 shows that the late language regions process information that is 1) partially correlated with low-level features, but also 2) partially independent of low-level features. This previous work shows that pretrained speech models align with late language regions because of 1).
> The inspiration behind our brain-tuning approach is to allow speech models to learn 2) directly from the brain signal. If successful, this approach would align with both 1) and 2) types of information in late language regions, thereby reducing the fraction of the alignment that is due only to 1) , i.e. the low-level impact on the brain alignment with late language regions. We will clarify this line of reasoning in the paper.
>
> 6. **Methodology figure X-axis labels**: We have added X-axis labels to Figure 1c to disambiguate the targeted downstream tasks by mentioning examples of tasks we expect these models to be measured on.
>
> We thank you very much for your thorough review! We hope we have addressed your questions and look forward to continuing the discussion if there are unresolved issues.
>
> [1] J. S. Garofolo, L. F. Lamel, W. M. Fisher, J. G. Fiscus, D. S. Pallett, and N. L. Dahlgren. The DARPA TIMIT Acoustic-Phonetic Continuous Speech Corpus CDROM. Linguistic Data Consortium, 1993.

---

> ### Comment · Reviewer_x8dp · 2024-11-25
>
> I thank the authors for taking the time respond. I would like to clarify a few things. Is this the procedure for brain-tuning?
> 1. An off-the-shelf speech model is taken
> 2. This model is fine-tuned to predict brain voxels and nothing else
> 3. When evaluated on speech datasets, the performance of the original speech model on audio tasks is not only retained, it actually improves.
>
> I feel I must be missing something. How can a model be trained on a completely different task, albeit one with correlation to the original task, and still retrain such good performance? I think it would be really good to test on the original data that wav2vec2 and HuBERT tested on, even if it is "biased" to the pretrained models. This will help determine what braintuning does to the original model performance.

---

> > ### Comment · Reviewer_x8dp · 2024-11-26
> >
> > A quick follow up question to the above.
> > - Lines 277-283 ("Downstream Evaluation") describe the finetuning process. On lines 282-283: it says that for ASR, the whole transformer is fine-tuned when training the new linear head $f$. My question is: is this procedure applied for the pretrained models as well? Or are the pretrained models taken off-the-shelf and evaluated?

---

> > > ### Author Response · Authors · 2024-11-26
> > >
> > > Thanks for engaging in a discussion. The understanding of the brain-tuning procedure is correct, except for 3: not all speech model performance improves. We observe that the low-level speech capabilities are not substantially changed by brain-tuning (Fig 11). We only observe an improvement in a number of more semantic tasks (Fig 4). Notably, the tasks where the brain-tuning is most helpful (the largest gap between brain-tuned and other baselines), are also the tasks where the new added baseline of fine-tuning using LM representations is helpful, further supporting the conclusion that the added benefits of brain-tuning are coming from improved semantics (Fig 10).
> > >
> > > Regarding the question about “how a model can be trained on a completely different task, albeit one with correlation to the original task, and still retain such good performance”? The point is that the “original task” that models are pretrained for are **not** the same as the downstream tasks that we test performance on. So we are not making any claims about the change in performance of models on their original pretraining tasks. We are making claims about their abilities to generalize to downstream tasks, which is a desirable property of a pretrained foundational model. We will clarify this in the paper. We hope this clarifies the reviewer's concern.
> > >
> > > Regarding evaluation on ASR: we aim to make all comparisons fair so we evaluate all models in the main paper (pretrained, BigSLM-finetuned, and brain-tuned) on ASR in the same way by first fine-tuning each model on the training portion of the ASR dataset. We will clarify this in the paper.

---

> > > > ### Comment · Reviewer_x8dp · 2024-11-26
> > > >
> > > > Thanks for the response!
> > > >
> > > > > The point is that the “original task” that models are pretrained for are not the same as the downstream tasks that we test performance on.
> > > >
> > > > Sorry, just to be on the same page, I am most interested in Figure 4a. Aren't wav2vec2 and HuBERT intended to be used for ASR? So isn't the claim that brain-tuning makes these models better at a task that they were originally trained for?
> > > >
> > > > > we evaluate all models in the main paper (pretrained, BigSLM-finetuned, and brain-tuned) on ASR in the same way
> > > >
> > > > Thanks. This is helpful for my understanding.

---

> > > > > ### Author Response · Authors · 2024-11-26
> > > > >
> > > > > Thank you for your continued engagement in the discussion and quick response.
> > > > >
> > > > > We are happy to clarify here and will also do so in the paper: Wav2vec2.0 and HuBERT are pretrained via self-supervised objectives that aim to predict representations of masked portions of the input audio. They are not pretrained on ASR. In order to perform ASR, the accepted evaluation strategy in the literature is to additionally fine-tune using text-audio paired data [1,2]. We perform this ASR-specific fine-tuning in the same way for all evaluated models, as clarified in the previous response. To be extra clear, the evaluation pipelines for ASR for each model that we have in the main paper are:
> > > > > - **Pretrained:** pretrained model -> fine-tune on ASR
> > > > > - **BigSLM fine-tuned:** pretrained model -> fine-tune using BigSLM representations -> fine-tune on ASR
> > > > > - **Brain-tuned:** pretrained model -> fine-tune using fMRI signals -> fine-tune on ASR
> > > > >
> > > > > In this way, all models are fine-tuned on ASR at the last stage right before the ASR evaluation which ensures fairness across models.
> > > > >
> > > > >
> > > > >
> > > > > [1] Baevski, A., Zhou, Y., Mohamed, A., & Auli, M. (2020). wav2vec 2.0: A framework for self-supervised learning of speech representations. Advances in neural information processing systems, 33, 12449-12460.
> > > > >
> > > > > [2] Hsu, W. N., Bolte, B., Tsai, Y. H. H., Lakhotia, K., Salakhutdinov, R., & Mohamed, A. (2021). Hubert: Self-supervised speech representation learning by masked prediction of hidden units. IEEE/ACM transactions on audio, speech, and language processing, 29, 3451-3460.

---

> > > > > > ### Comment · Reviewer_x8dp · 2024-11-27
> > > > > >
> > > > > > I thank the authors for their detailed replies. I understand the intent behind their ASR experiments better now.
> > > > > >
> > > > > > I think that it is important to be be careful with the claim that brain-tuning improves ASR! The gold standard for showing an improvement on a task is to show stronger numbers on the same data that the past state-state-of-the-art used to evaluate, i.e. LibreSpeech. This is the established practice for most machine learning tasks, e.g. object detection, ASR, etc. For this reason, I do not think the experiments can support the claim that brain-tuning shows "*consistent improvements in performance on ... downstream tasks*". Instead, a more supportable claim could be "*for the datasets tested, brain-tuning was found to be a useful additional pre-training task*". Or some other phrasing that did not imply a definitive improvement over the SOTA. Or maybe this could be discussed in a limitations section?
> > > > > >
> > > > > > In general, I think this is an interesting paper, and that it is hovering right on the accept border. I will be happy to increase my score if the authors can find a way to narrow the claims about downstream improvement or otherwise state some sort of caveat.

---

> > > > > > > ### Author Response · Authors · 2024-11-28
> > > > > > >
> > > > > > > We thank the reviewer for the continued engagement. Our intention is not to claim SOTA on any one particular task but to provide evidence for our central claim (also stated in the title) that brain-tuning improves semantic understanding in speech models, which we believe our experiments indeed show. We've tried to be careful about stating this in the paper but see now that there are several points in the paper where this can be clarified -- thank you for pointing this out.
> > > > > > >
> > > > > > > We've now revised the paper to make this more clear in 3 main ways:
> > > > > > > 1) qualifying more general language about improvements in performance on “downstream tasks” to "downstream **semantic** tasks" (e.g. in the end of the abstract),
> > > > > > > 2) included a discussion in Section 3.4.3 (and Appendix D.2) about how we now utilize the LM-finetuned baseline to provide empirical evidence on which tasks we should expect semantics to be helpful for,
> > > > > > > 3) revised the discussion of the downstream task results in Section 4.3 to specifically say that the tasks for which brain-tuning is most helpful (i.e. the improvement above the BigSLM baseline is the largest) are the ones where we also find that the LM-finetuned baseline is also helpful. Note that ASR is not one of these tasks as the LM-finetuned baseline does not outperform the BigSLM baseline on it (updated Fig 10 with ASR results), and the brain-tuned model performs on par with the BigSLM baseline (Fig 4 and 10).
> > > > > > >
> > > > > > > In summary, we are now more clear about the tasks that we expect brain-tuning to be helpful for (i.e. more semantic tasks) and focus the discussion of results on these tasks. For the remaining tasks, we now discuss how brain-tuning does not negatively impact performance (unlike LM-finetuning, Fig 10).
> > > > > > >
> > > > > > > We hope this discussion alongside the revised manuscript addresses your concerns, and we thank you for helping us communicate our work more clearly.

---

> > > > > > > > ### Comment · Reviewer_x8dp · 2024-12-01
> > > > > > > >
> > > > > > > > I thank the authors for their continued discussion. I have upgraded my score.

---

> > > > > > > > > ### Author Response · Authors · 2024-12-03
> > > > > > > > >
> > > > > > > > > We thank the reviewer for the responsiveness and valuable insights, and for showing support by upgrading the score.

---

### Author Response · Authors · 2024-11-23

We thank all reviewers for their detailed and constructive feedback. We were very happy to see that the reviewers believe our work is "noteworthy” (x8dp, BDuH), "beneficial and impactful” (BDuH), "novel” and "thorough” (SHAj), “impressive” and “very well written” (61Sy).

The reviews highlight two main areas that can help strengthen the confidence in the results: adding a new baseline and adding new results on additional low-level speech tasks. We do both in our response and show that our brain-tuned models behave as we expect: outperform a new language model-finetuned baseline, and do not significantly impact performance on low-level speech tasks. We provide more details below in the common response as well as respond to each individual reviewer concerns in separate threads. We additionally include some clarifications that we believe are helpful:

1. **ASR results in Fig 4a:** The results are correct and were correctly reported in the text, but the plot in the submission showed the word error rate for two of the models, rather than the accuracy (1-word error rate). We have now fixed this issue in the revised manuscript (Fig 4a). We understand how this subplot was confusing and apologize for this error. We hope that the revised figure alleviates the concerns regarding what appeared to be a low pretrained performance on ASR. In fact, the pretrained ASR performance we report on the TIMIT dataset is on par with previous results reported on this dataset. [1, 2]. The brain-tuned model outperforms the pretrained model by 12%, as reported in the text.

2. **Datasets for downstream evaluation:** The datasets used for evaluation on the downstream tasks are popular benchmark datasets and are completely independent of the fMRI data that the brain-tuned model had access to. The details of these datasets can be found in Section 3.4.3 and Appendix B. We believe that using datasets that are new to all models leads to more fair evaluation and increases confidence in the downstream results.

3. **New results on additional low-level speech tasks:** To alleviate Reviewers x8dp and BDuH's concerns about potential catastrophic forgetting of speech capabilities, we now evaluate additional low-level speech tasks: MFCC and FBANK prediction. We find that brain-tuned and BigSLM models perform on par with the pretrained model on these low-level tasks, indicating that brain-tuning does not significantly impact low-level speech capabilities (Appendix D.1). We believe this is due to the fact that the brain regions that are included in the training do not only include semantic regions but also low-level speech regions (the auditory cortex). Moreover, we observe that the model layers that are affected most by brain-tuning are middle to late layers (Appendix D.1), and not the early layers which best encode the core acoustic features [3].

4. **New baseline and clarification on existing baselines:**

    1. **Language Model-finetuned Baseline**: Reviewer BDuH suggested a new baseline which uses representations from a Language Model (LM) as a fine-tuning target. We implemented an LM-finetuned baseline using representations from GPT2, and we report its downstream performance in Appendix D.2. Most notably, for all reported tasks, the brain-tuned model outperforms the GPT2-finetuned baseline. As compared to the BigSLM-finetuned models, the GPT2-finetuned baseline falls behind it in 4 of the 6 reported tasks. These results don’t show evidence that an LM-finetuned baseline would necessarily be consistently better than a BigSLM-finetuned baseline.

    2. **BigSLM baseline**: We believe this baseline, which uses representations from a large speech model as a fine-tuning target, is the best comparison to test the added value of brain data over speech models, because: 1) It shows what semantics the fMRI targets provide that representations from an advanced speech model do not. 2) It allows us to contrast how the speech-related abilities of the models are affected (decreased or increased) by fMRI targets as compared to targets from an advanced speech model. This intuition is confirmed by BigSLM fine-tuned models generally performing better or on par with the pretrained ones on downstream tasks (especially ASR).

    3. **Random-fMRI baseline**: This baseline is useful for two reasons. First, it ensures that having the correctly aligned fMRI target is essential (i.e., not any listening brain signal can work). Second, noise injection and noisy targets can act as regularization and have positive effects on the model [4, 5]. Hence, grounding results obtained with correct brain targets by comparing them to what random targets can achieve is necessary to ensure there are benefits to the process beyond a simple regularization effect.

We believe the new results we have provided have strengthened our work and hope that they address the concerns of the reviewers. We will appreciate the opportunity to engage in a discussion if there are remaining concerns.

---

> ### Author Response · Authors · 2024-11-23
>
> [1] A. Baevski, W. Hsu, A. Conneau, and M. Auli. "Unsupervised speech recognition." NeurIPS 2021.
>
> [2] https://github.com/elgeish/transformers/tree/cfc0bd01f2ac2ea3a5acc578ef2e204bf4304de7
>
> [3] A. Vaidya, S. Jain, and A. Huth. Self-supervised models of audio effectively explain human cortical responses to speech. ICML 2022.
>
> [4] S. Zada, I. Benou, and M. Irani. "Pure noise to the rescue of insufficient data: Improving imbalanced classification by training on random noise images." ICML 2022.
>
> [5] C. Bishop, "Training with Noise is Equivalent to Tikhonov Regularization," in Neural Computation 1995.

---

> > ### Author Response · Authors · 2024-12-03
> >
> > We thank all reviewers for the continued and insightful discussions. We have made several additions to the paper in light of these discussions, which we believe increase the clarity and strength of our work. We summarize these additions and their purpose below:
> > 1. **Adding LM-finetuned Baseline** [Reviewers BDuH and SHAj]: To provide a semantically rich comparison model, we implemented an LM-finetuned baseline using representations from GPT2, and we report its downstream performance on all tasks in App. D.2. We see that the tasks in which the LM helps most are the same tasks in which the brain-tuned models substantially outperform pretrained and BigSLM-finetuned models, further supporting our claim that brain-tuning improves semantic understanding in speech models.
> > 2. **Focusing Discussion of Downstream Benefit of Brain-tuning  on Semantic Tasks** [Reviewers x8dp and BDuH]: We now clarify our evaluation methods and provide further evidence that brain-tuning helps most in semantic tasks by: 1) utilizing the LM-finetuned baseline to provide empirical evidence on which tasks we should expect semantics to be helpful for, and 2) revising the paper sections on downstream task to specifically highlight our focus on semantic tasks and that the tasks for which brain-tuning is most helpful are the ones where we also find that the LM-finetuned baseline is also helpful.
> >
> > 3. **New Experiments on Effect of Data Size and Model Size on Brain-tuning** [Reviewer 61Sy]: We carried out experiments showing the gain in performance of brain-tuned models relative to pretrained models when we increase the model size and when we increase the data size (App. E.3). Our results provide clear evidence that data size plays a key role in the positive results we see from brain-tuning. Moreover, the increasing trend in performance has not leveled out yet, suggesting that incorporating more brain data during brain-tuning will yield further benefits. Since there is no current dataset with more fMRI data per participant, future methods may work to combine data from multiple participants or multiple experiments to provide further gains.
> >
> > 4. **Performance on Low-level Tasks** [Reviewers x8dp and BDuH]: To show that no catastrophic forgetting happens to the low-level performance of the brain-tuned models, we reported the low-level performance compared to the pretrained, BigSLM-finetuned and LM-finetuned models (App. D.1 & D.2). We show that, unlike the LM-finetuned baseline, Brain-tuning doesn’t degrade the low-level performance substantially at the cost of adding semantics. This is a crucial result to prove that brain-tuning doesn’t compromise the utility of the speech model.
> >
> >
> > Lastly, we summarize below the additions to be included in the final version of the paper if it gets accepted:
> >
> > 1. As promised in the very recent discussion with Reviewer SHAj, we will work to include two other baselines: an LLama2 7b fine-tuned baseline and an fMRI-stimulus audio pretraining baseline in the final version of the paper. We do not expect these results to change any of our key takeaways, as we discuss in the individual response to reviewer SHAj.
> > 2. As promised in our very recent reply to Reviewer BDuH, we will add more discussion on what the fMRI signals can improve and maintain in the speech models to the introduction and related work sections of the final version of the paper.
> >
> > We thank all reviewers again for the valuable discussions; they helped us clarify our message and provide even stronger evidence for our main takeaways!

---

### Meta-Review · Area_Chair_vT86 · 2024-12-21

**Metareview:**

The paper introduces "brain-tuning," a method for fine-tuning speech language models (SLMs) using fMRI data from individuals listening to natural stories. The central claim is that incorporating brain signals into the training process enhances the models' semantic understanding, making them better aligned with how the human brain processes language. The work has implications for both cognitive neuroscience (understanding semantic processing in the brain) and AI (developing more human-like language models). The reliance on fMRI data is a practical limitation. While not a weakness of the paper's core idea, the reliance on fMRI data, which is expensive and difficult to acquire, could limit the scalability of the approach. The authors acknowledge this and suggest future work on combining data from multiple participants or experiments. Initial claims of "consistent improvements" on downstream tasks were too broad. Reviewers raised concerns about the ASR results, leading the authors to clarify that the main benefits are observed in more semantic tasks. Reviewer SHAj suggested adding a baseline pretrained on X + the fMRI stimulus audio, which the authors agreed to include in the final version. This will provide a crucial control for the effect of the added audio data.

**Additional Comments On Reviewer Discussion:**

The authors have been highly responsive to reviewer feedback, conducting new experiments (e.g., LM-finetuned baseline, low-level speech tasks, effects of model and data size) and providing clarifications that strengthen the paper.

---

### Decision · Program_Chairs · 2025-01-22

Accept (Poster)